# Revisiting Maximum Mean Discrepancy via Diffusion Behavior Policy in Offline RL: A Mode-Seeking Perspective

## Abstract

Policy constraint is an effective way to mitigate distributional shift in offline Reinforcement Learning (RL). However, a key challenge lies in identifying the mode of the behavior distribution that corresponds to the highest return, thereby avoiding unnecessary constraints on suboptimal actions. The reverse KL divergence constraint can provide this capability, but its efficacy is limited by the fidelity of the behavior model, such as Gaussian distributions. Diffusion models, while providing expressive behavior, cannot be directly employed with KL divergence due to the absence of analytic probability formula. In contrast, the Maximum Mean Discrepancy (MMD) constraint operates solely on samples generated by diffusion policies, prompting us to re-examine its potential. Surprisingly, our numerical studies reveal an intriguing insight: MMD exhibits strong mode-seeking capabilities when applied to high-fidelity diffusion behavior policies guided by value signals. This finding corrects a misunderstanding in previous MMD-based methods and shows that their failures were primarily due to distorted behavior modeling. We further investigate the effect of value perturbation on MMD's mode-seeking behavior and, accordingly, revolutionize the MMD-based policy constraint method for offline RL. Extensive experiments on the D4RL benchmark show that our method significantly outperforms prior MMD-based methods and achieves state-of-the-art performance.

## 1 Introduction

Although Reinforcement Learning (RL) (Sutton & Barto, 2018) has achieved remarkable success in games (Mnih et al., 2015; Vinyals et al., 2019), its paradigm of online trial-and-error learning has become a barrier to practical applications (Kiran et al., 2022). Offline reinforcement learning (Fujimoto et al., 2019; Levine et al., 2020), which relies solely on offline collected datasets without requiring online interaction, shows potential in enhancing the generalization of RL models and their applicability in real-world scenarios. While there is considerable promise associated with this approach, existing research indicates that offline RL faces the challenge of distributional shift (Levine et al., 2020), making policy evaluation difficult and, consequently, hindering policy optimization.

Recent advancements in policy constraint methods for offline reinforcement learning have introduced a variety of strategies to address the challenge of distributional shift. BCQ (Fujimoto et al., 2019) applies action perturbation to the behavior policy modeled by a CVAE (Kingma & Welling, 2022), thereby constraining the learned policy. BRAC (Wu et al., 2019) employs KL divergence to restrict the policy within the trust region of the behavior policy. TD3+BC (Fujimoto & Gu, 2021) directly uses a Mean Squared Error (MSE) loss to regulate the policy, while approaches such as DOGE (Li et al., 2023) and OAP (Yang et al., 2023) build on this method by incorporating action-selective weights into the MSE loss. More recently, Diffusion-QL (DQL) (Wang et al., 2023) and DTQL (Chen et al., 2024c) have adopted denoising loss to effectively enforce policy constraints.

Despite these advancements, a critical question remains open and thought-provoking: *how to effectively identify the support of the behavior policy's distribution that aligns with the maximum return, thereby avoiding overly constraints on suboptimal regions.* Although reverse KL divergence has strong support matching capabilities (Wu et al., 2019; Cai et al., 2022), its effectiveness is hindered

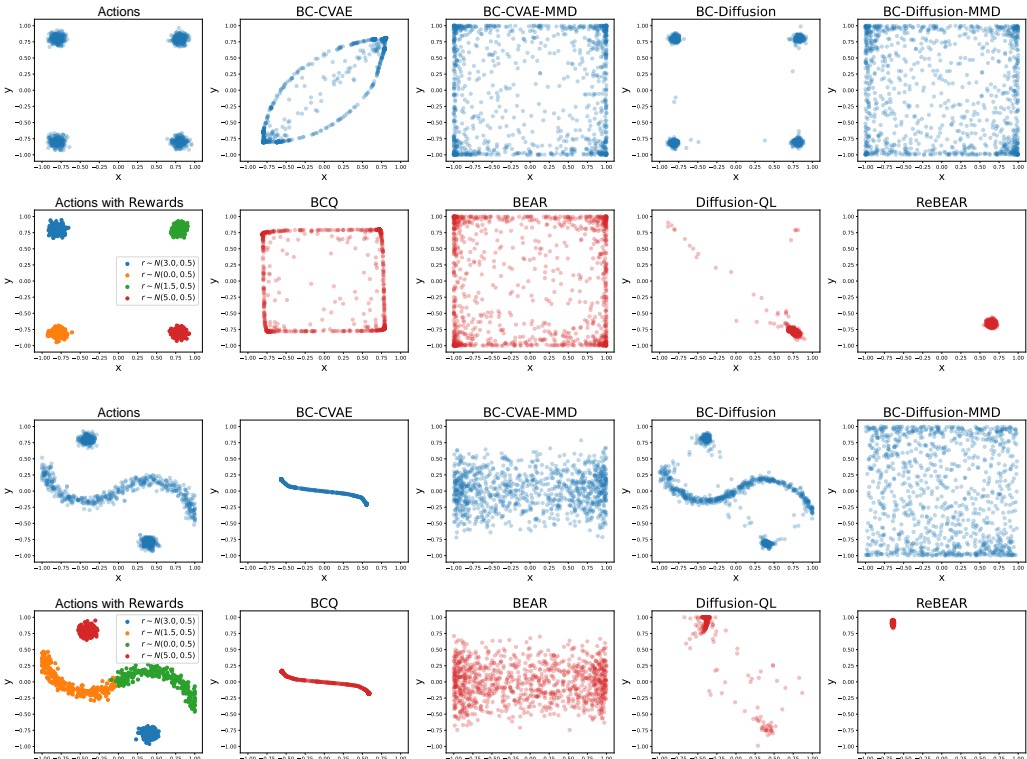

Figure 1: Two 2D bandit examples illustrate MMD's mode-seeking ability under **accurate behavior modeling** and **value guidance**. Points denote actions, and colors in the first column indicate random rewards. Results of 'BC-Diffusion-MMD' show that direct policy cloning under MMD fails to align with the support, despite using the high-fidelity behavior diffusion policy. Surprisingly, ReBEAR demonstrates impressive mode-seeking capability when combining 'BC-Diffusion-MMD' with Q-value, outperforming Diffusion-QL, which suffers from "tail shadow" effects—i.e., still sampling a few suboptimal actions. The previous MMD-based method, BEAR, fails despite incorporating Q-value maximization with MMD constraints, likely due to blurred behavior modeling by the CVAE. We provide theoretical analysis of this success behind 'ReBEAR' in Section 3.4.

by the limitations of behavior modeling, such as assumptions of Gaussian distributions. While diffusion models offer highly expressive behavior distributions (Ho et al., 2020), the lack of explicit probabilistic formula renders them incompatible with KL divergence-based constraints.

In this work, we revisit Maximum Mean Discrepancy (MMD) as a policy constraint, leveraging its ability to operate solely on samples generated by diffusion policies. To validate its potential for mode-seeking, we conduct experiments on two 2D bandit examples, as illustrated in Fig. 1. These experiments reveal two counter-intuitive phenomena. First, *directly cloning a Gaussian distribution from a high-fidelity diffusion policy using MMD fails to exhibit mode-seeking capability* ('BC-Diffusion-MMD' in Fig. 1), which can easily lead to misleading conclusions. More intriguingly, the result of 'ReBEAR' in Fig. 1 highlights *the precise mode-seeking ability of MMD when combined with the Q-value (expected reward in bandit) and the diffusion behavior policy*, even surpassing the Diffusion-QL (DQL). These suggest that one of the reasons behind the poor performance of BEAR (Kumar et al., 2019), previous MMD-based method, may stem from *the distorted behavior modeling by CVAE*. Additionally, we will identify a secondary failure reason of BEAR related to value estimation, which is analyzed in detail in the method section and our ablation studies.

To address these limitations, we revolutionize the MMD-based constraint method and respectfully name our method ReBEAR (Revisiting BEAR) in recognition of the original BEAR framework. Experiments demonstrate that ReBEAR achieves a remarkable 103.2% improvement across Gym tasks, a 255.7% improvement across AntMaze, and consistent superiority on Adroit compared to BEAR. In particular, our method *corrects a long-standing misunderstanding* about the mode-seeking

property of MMD-based constraints. Moreover, ReBEAR can be viewed as *distilling a Gaussian policy from the expressive diffusion-based behavior policy via MMD*, which not only preserves the fidelity of the diffusion model but also *enables efficient inference* through the Gaussian model.

## 2 PRELIMINARIES

RL is typically formulated as a Markov Decision Process (MDP) (Sutton & Barto, 2018), denoted as $\langle \mathcal{S}, \mathcal{A}, P, \rho_0, r, \gamma \rangle$, where $\mathcal{S}$ represents the state space, $\mathcal{A}$ the action space, $P(\cdot \mid s, a)$ the transition probability function, $\rho_0(\cdot)$ the initial state distribution, $r(s, a)$ the reward function, and $\gamma \in (0, 1)$ the discount factor. The objective of an RL agent is to learn a policy $\pi(\cdot \mid s)$ (Gaussian in our case) that maximizes the expected cumulative discounted rewards (return). The Q-value, $Q(s, a)$, represents the expected return when starting from state $s$ and taking action $a$ under the policy being evaluated. To estimate the Q-value, the Bellman operator is commonly employed:

$$(\mathcal{T}^\pi Q)(s, a) := r(s, a) + \gamma \mathbb{E}_{s' \sim P(\cdot|s,a), a' \sim \pi(\cdot|s')} \left[ Q(s', a') \right], \tag{1}$$

and then the estimation of the Q-value is updated by minimizing the mean squared Bellman error:

$$\mathbb{E}_{(s,a,r,s')} \left[ (Q(s, a) - \mathcal{T}^\pi Q(s, a))^2 \right]. \tag{2}$$

In offline RL (Levine et al., 2020; Prudencio et al., 2023), the agent is restricted to learning purely from a fixed dataset of transitions, $\mathcal{D} = \{(s, a, r, s')\}$, collected under an unknown behavior policy $\pi_\beta(\cdot \mid s)$. While the objective remains the same as in online RL—maximizing cumulative rewards—classical methods often struggle in this setting due to distributional shift (Levine et al., 2020). To address this challenge, various strategies have been proposed, including value regularization (Kumar et al., 2020) and policy constraints (Fujimoto et al., 2019).

### 2.1 BEAR

BEAR (Kumar et al., 2019) is one of the policy constraint methods, which keeps policy evaluation as in TD3 (Fujimoto et al., 2018) while restrict the learning policy using MMD as:

$$\pi_\phi := \max_\pi \mathbb{E}_{s \sim \mathcal{D}} \mathbb{E}_{a \sim \pi(\cdot|s)} \left[ \min_{j=1,\ldots,K} Q_j(s, a) \right] \tag{3}$$

$$\text{s.t. } \mathbb{E}_{s \sim \mathcal{D}}[\text{MMD}(\pi_\beta(\cdot \mid s), \pi(\cdot \mid s))] \leq \varepsilon$$

where $K$ is number of critic networks, $\varepsilon$ is an approximately chosen threshold hyperparameter. In practice, the sampled version of MMD (Gretton et al., 2012) is implemented as follows

$$\text{MMD}^2 \left( \mathcal{D}_1(\cdot), \mathcal{D}_2(\cdot) \right) := \text{MMD}^2 \left( \{x_1, \cdots, x_n\}, \{y_1, \cdots, y_m\} \right)$$

$$= \frac{1}{n^2} \sum_{i,i'} k(x_i, x_{i'}) - \frac{2}{nm} \sum_{i,j} k(x_i, y_j) + \frac{1}{m^2} \sum_{j,j'} k(y_j, y_{j'}) \tag{4}$$

where $x_1, \cdots, x_n \sim \mathcal{D}_1$ and $y_1, \cdots, y_m \sim \mathcal{D}_2$ are corresponding samples, $m$ and $n$ are number of samples, $k(\cdot, \cdot)$ can be Laplacian or Gaussian kernel with bandwidth parameter $\sigma$.

## 3 METHOD

In this section, we propose two simple yet key enhancements to address the limitations of BEAR. As observed in the 2D bandit examples (Fig. 1), accurate behavior modeling plays a pivotal role in learning support-constrained policies using MMD. To this end, we first integrate diffusion models for precise behavior policy modeling. Subsequently, we identify another potential factor contributing to BEAR's shortcomings and introduce an additional value penalty, detailed in the second subsection. Finally, we provide a theoretical analysis of the mode-seeking phenomenon of the MMD constraint.

### 3.1 BEHAVIOR MODELING WITH DIFFUSION MODEL

In practice, the behavior policy $\pi_\beta$ required in (3) is typically unknown. BEAR addresses this by employing a CVAE model to approximate $\pi_\beta$. However, as demonstrated in the 2D bandit examples,

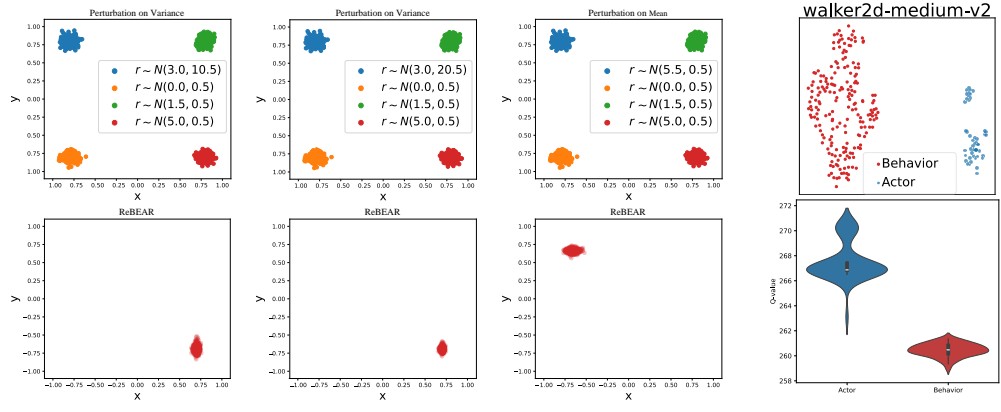

(a) Reward variance 10.5 (b) Reward variance 20.5 (c) Mean shift to 5.5 (d) OOD overestimation

Figure 2: Despite noisy rewards making suboptimal actions (blue) occasionally appear optimal in (a)-(b), the MMD constraint reliably matches true optimal actions. In (c), artificially increasing suboptimal actions' reward beyond the optimal ones causes rapid policy support shift. This reveals that MMD constraint maintains robust mode-seeking while remaining sensitive to Q-estimation bias. (d) The t-SNE visualization on walker2d-medium without value penalty. For 100 random states, 50 actions from the learned policy and 200 from the behavior policy are sampled. In 88% of state-cases, policy actions lie outside the behavior support with overestimated values.

CVAE tends to blur the support of the behavior policy, resulting in BEAR's inability to effectively control the learned policy. In contrast, combining accurate behavior policy modeling with MMD exhibits strong mode-seeking capabilities.

Motivated by this insight, we model $\pi_\beta(a \mid s)$ using a diffusion model (denoted as $\mathrm{Diff}_\omega(a \mid s)$) to preserve the behavior support. Specifically, it is constructed via a reverse diffusion chain, formulated as $\mathcal{N}(a^L; 0, I) \prod_{l=1}^{L} p_\omega(a^{l-1} \mid a^l, s)$, where $l$ denotes the diffusion timestep, $a := a^0$ is the final sampled action, $a^l$, $l = 1, \cdots, L-1$, are latent variables, $a^L \sim \mathcal{N}(0, I)$ is the Gaussian noise. Typically, $p_\omega(a^{l-1} \mid a^l, s)$ is modeled as a Gaussian distribution $\mathcal{N}\left(a^{l-1}; \mu_\omega(a^l, s, l), \Sigma_\omega(a^l, s, l)\right)$ with the covariance matrix $\Sigma_\omega(a^l, s, l) = \beta_l I$ and the mean defined as

$$\mu_\omega(a^l, s, l) = \frac{1}{\sqrt{\alpha_l}} \left( a^l - \frac{\beta_l}{\sqrt{1 - \bar{\alpha}_l}} \xi_\omega(a^l, s, l) \right), \tag{5}$$

where $\beta_l$ is the variance schedule, $\alpha_l := 1 - \beta_l$, $\bar{\alpha}_l := \prod_{i=1}^{l} \alpha_i$, and $\xi_\omega(\cdot)$ is the noise prediction network with parameters $\omega$. The conditional diffusion model is optimized by maximizing the evidence lower bound, which can be simplified (Ho et al., 2020) to minimize the following objective

$$\min_\omega \mathbb{E}_{\substack{l \sim \mathcal{U}(\cdot), (s,a) \sim \mathcal{D} \\ \xi \sim \mathcal{N}(0, I)}} \left\| \xi - \xi_\omega(\sqrt{\bar{\alpha}_l} a + \sqrt{1 - \bar{\alpha}_l} \xi, s, l) \right\|^2, \tag{6}$$

where $\mathcal{U}(\cdot)$ is an uniform distribution over $\{1, \cdots, L\}$.

### 3.2 TARGET Q-VALUE PENALTY

For most tasks in the D4RL benchmark (Fu et al., 2020), performances are significantly improved after adopting the behavior diffusion policy under MMD. However, a few tasks, such as walker2d-medium-replay and walker2d-medium, still exhibit suboptimal performance, as will be demonstrated in our ablation studies. Notably, Fig. 1 implicitly assumes accurate Q-value estimation. These observations motivate us to further investigate how value perturbations influence the mode-seeking capability under MMD.

Figure 2(a) and (b) illustrate results when introducing varying levels of noise to the reward of suboptimal actions (blue points), causing them to occasionally receive the highest reward. Remarkably, Fig. 2 demonstrates that the MMD constraint exhibits a *robust mode-seeking ability*, consistently identifying the optimal action, even when the variance reaches 20.5. However, when the mean of reward is artificially increased to 5.5 shown in Fig. 2(c)—slightly higher than the optimal action's reward of 5.0—the learned policy rapidly shifts its focus to the support of these blue-sample points.

This suggests that MMD's strong mode-seeking capability is a double-edged sword: *when overly optimistic value estimates arise for suboptimal actions, the algorithm quickly adapts to them, potentially leading to suboptimal solutions*. We believe this phenomenon underlies the failures observed in tasks, such as walker2d-medium. To mitigate this, we introduce a penalty on out-of-distribution actions when computing target Q-values. For simplicity, we also adopt MMD distance as the penalty term. Specifically, we replace the Bellman operator in Eq. (2) with a modified operator as follows:

$$
\begin{aligned}
(\mathcal{T}_{\mathrm{MMD}}^{\pi} Q)(s, a) := r(s, a) + \gamma \mathbb{E}_{s' \sim P(\cdot|s,a), a' \sim \pi(\cdot|s')} \Big[ & Q(s', a') \\
& - \beta_c \, \mathrm{MMD}(\mathrm{Diff}_\omega(\cdot \mid s'), \pi(\cdot \mid s')) \Big],
\end{aligned}
\tag{7}
$$

where $\beta_c$ is the penalty coefficient. While value penalty has shown negligible performance impact when applied to KL divergence (Wu et al., 2019) and MSE loss (Tarasov et al., 2023), we observe the opposite effect for MMD. Our ablation studies demonstrate the efficacy of this modification, which ultimately contributes to state-of-the-art performance.

### 3.3 BRIEF SUMMARY OF ReBEAR

In summary, ReBEAR follows an actor-critic learning framework. The critic network update parameters $\theta$ by minimizing the mean squared Bellman error with the modified operator in (7), as:

$$
\mathbb{E}_{(s,a,r,s') \sim \mathcal{D}} \left[ (Q_\theta(s, a) - \mathcal{T}_{\mathrm{MMD}}^{\pi_\phi} Q_{\theta^-}(s, a))^2 \right],
\tag{8}
$$

where $\theta^-$ is the parameters of target network. For policy improvement, ReBEAR employs an MMD-based policy constraint, as formulated in Eq. (3). In practice, this constraint is implemented as a regularization term and expressed as:

$$
\max_\phi \mathbb{E}_{s \sim \mathcal{D}} \mathbb{E}_{a \sim \pi_\phi(\cdot|s)} \left[ \min_{j=1,2} Q_{\theta_j}(s, a) \right] \\
- \beta_a \mathbb{E}_{s \sim \mathcal{D}} [\mathrm{MMD}(\mathrm{Diff}_\omega(\cdot \mid s), \pi_\phi(\cdot \mid s))],
\tag{9}
$$

where $\beta_a$ is the penalty coefficient for the actor, and $\theta_j$, $j = 1, 2$ means double Q-network is utilized. As mentioned before, $\mathrm{Diff}_\omega(\cdot \mid s)$ is the diffusion model for modeling behavior policy $\pi_\beta$ and learned by minimizing the diffusion loss in Eq. (6). The complete algorithm is provided in the Appendix.

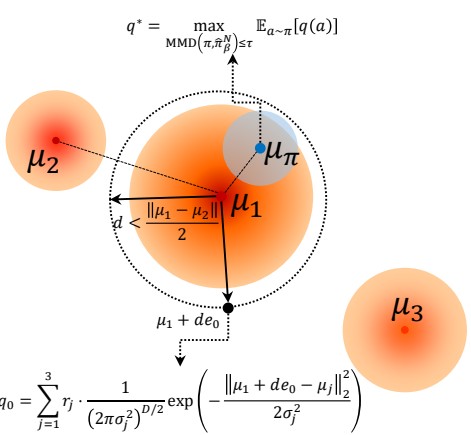

Figure 3: Assuming the rewards satisfy $r_1 > r_2 > r_3$, If one can obtain $q^* \geq q_0$ under some conditions, $\|\mu_\pi - \mu_1\|$ is then expected to less than $d$. Therefore, mode-seeking property is demonstrated.

### 3.4 THEORETICAL ANALYSIS

We provide mathematical analysis of this success in the bandit example. To facilitate our analysis, a schematic is illustrated in Fig. 3. In this context, the behavior policy $\pi_\beta$ is a Gaussian mixture distribution composed of $M$ Gaussian policies: $\pi_{\beta_j} \sim \mathcal{N}(a \in \mathbb{R}^D \mid \mu_j, \sigma_j^2 I)$, $j = 1, \cdots, M$. Assume each action sampled from the $j$-th mode policy $\pi_{\beta_j}$ yields a reward $r_j$, with rewards ordered as $r_1 > r_2 > \cdots > r_M$. Consequently, the expected action-value for any action $a$ is given by $q(a) = \sum_{j=1}^M \pi_{\beta_j}(a) r_j$. Under the MMD constraint, the optimal value is obtained by solving:

$$
q^* = \max_{\mathrm{MMD}(\pi, \widehat{\pi}_\beta^N) \leq \tau} \mathbb{E}_{a \sim \pi}[q(a)],
\tag{10}
$$

where $\tau$ controls the tightness of the policy constraint, and $\widehat{\pi}_\beta^N = \frac{1}{N} \sum_{i=1}^N \delta_{a_i}$ is the empirical behavior distribution, $\delta_{a_i}$ denotes the Dirac delta function for action samples $a_i$, $i = 1, \cdots, N$, and $N$ is the sample size. For the action $\mu_1 + de_0$, where $\|e_0\|_2 = 1$ and $d$ is a constant denoting distance, its expected return takes the form:

$$
q_0 = \sum_{j=1}^M r_j \cdot \frac{1}{(2\pi\sigma_j^2)^{D/2}} \exp\left(-\frac{\|\mu_1 + de_0 - \mu_j\|_2^2}{2\sigma_j^2}\right).
\tag{11}
$$

Table 1: Performance comparison on MuJoCo '-v2' tasks. The results represent the mean and standard error of normalized scores over 5 random seeds (each evaluated with 10 trajectories). The abbreviations are: m = medium, mr = medium-replay, and me = medium-expert.

| Task Name | BC | BCQ | CQL | IQL | TD3+BC | DQL | OAP | DTQL | SRPO | BEAR | ReBEAR(Ours) |
|---|---|---|---|---|---|---|---|---|---|---|---|
| halfcheetah-m | 42.6 | 46.6 | 44.0 | 47.4 | 48.3 | 51.1 | 56.4 | 57.9 | **60.4** | 41.0 | **66.6** ± 0.7 |
| hopper-m | 52.9 | 59.4 | 58.5 | 66.3 | 59.3 | 90.5 | 82.0 | **99.6** | 95.5 | 51.9 | **98.8** ± 1.6 |
| walker2d-m | 75.6 | 71.8 | 72.5 | 78.3 | 83.7 | 87.0 | 85.6 | **89.4** | 84.4 | 80.9 | **90.5** ± 0.8 |
| halfcheetah-mr | 36.3 | 42.2 | 45.5 | 44.2 | 44.6 | 47.8 | **53.4** | 50.9 | 51.4 | 29.7 | **52.3** ± 0.7 |
| hopper-mr | 18.1 | 60.9 | 95.0 | 94.7 | 60.9 | **101.3** | 98.5 | 100.0 | 101.2 | 37.3 | **102.8** ± 0.2 |
| walker2d-mr | 26.0 | 57.0 | 77.2 | 73.9 | 81.8 | **95.5** | 84.3 | 88.5 | 84.6 | 18.5 | **98.4** ± 1.3 |
| halfcheetah-me | 55.2 | 95.4 | 91.6 | 86.7 | 90.7 | **96.8** | 83.4 | 92.7 | 92.2 | 38.9 | **103.6** ± 1.3 |
| hopper-me | 52.5 | 106.9 | 105.4 | 91.5 | 98.0 | **111.1** | 85.9 | 109.3 | 100.1 | 17.7 | **112.3** ± 0.4 |
| walker2d-me | 101.9 | 107.7 | 108.8 | 109.6 | 110.1 | 110.1 | **111.1** | 110.0 | **114.0** | 95.4 | 110.5 ± 0.2 |
| Total score | 461.1 | 647.9 | 698.5 | 692.6 | 677.4 | 791.2 | 740.6 | 798.3 | 783.8 | 411.3 | **835.8** |

If we can prove that $q^* \geq q_0$ under appropriate conditions on $\tau$, it follows that $\mu_\pi$ will be close to $\mu_1$ when $d$ is sufficiently small. We formalize this result in the following theorem.

**Theorem 3.1.** *Let $\vec{q}$ be the vector with $i$-th element $q(a_i)$ and kernel matrix $K$ with its element $K_{ij} = k(a_i, a_j)$ used in the MMD constraint. When $\tau$ satisfies the following conditions:*

$$
\tau \geq \frac{q_0 - (\mathbf{1}^{\mathrm{T}} \vec{q})/N}{\sqrt{\vec{q}^{\mathrm{T}} K^{-1} \vec{q} - \frac{(\vec{q}^{\mathrm{T}} K^{-1} \mathbf{1})^2}{\mathbf{1}^{\mathrm{T}} K^{-1} \mathbf{1}}}} \ and \ \tau \cdot K^{-1} \left( \frac{\mathbf{1}^{\mathrm{T}} K^{-1} \vec{q}}{\mathbf{1}^{\mathrm{T}} K^{-1} \mathbf{1}} \mathbf{1} - \vec{q} \right) \preceq \frac{\sqrt{\vec{q}^{\mathrm{T}} K^{-1} \vec{q} - \frac{(\mathbf{1}^{\mathrm{T}} K^{-1} \vec{q})^2}{\mathbf{1}^{\mathrm{T}} K^{-1} \mathbf{1}}}}{N} \mathbf{1},
$$

(12)

*where $\mathbf{1}$ is a vector with all elements are one and $\preceq$ indicates element-wise inequality, we can indeed conclude that $q^* \geq q_0$.*

We provide a detailed proof of the theorem in the Appendix, along with a concrete example demonstrating that the required conditions can be readily satisfied.

## 4 EXPERIMENTS

In this section, we evaluate our method based on the standard D4RL benchmark (Fu et al., 2020), aiming to answer the following questions: 1) How much improvement does ReBEAR achieve over BEAR? How does ReBEAR compare to other state-of-the-art policy constraint methods? 2) How do key hyperparameters influence the performance? Is ReBEAR sensitive to these hyperparameters? 3) What impact does the penalty term on the target Q-value have on performance?

### 4.1 EXPERIMENTAL SETTINGS

We conduct comprehensive evaluations on MuJoCo, AntMaze and Adroit tasks from D4RL benchmark (Fu et al., 2020). Our main comparisons focus on policy constraint methods. Behavior Cloning (BC) results are referenced from DTQL (Chen et al., 2024c). For the classical value regularization method CQL (Kumar et al., 2020), we use results from IQL (Kostrikov et al., 2022), as the original study was conducted on '-v0' datasets. The performance of BCQ (Fujimoto et al., 2019) and BEAR (Kumar et al., 2019) is sourced from the official D4RL benchmark report (Fu et al., 2020). Results for other methods are taken from their respective original papers, including IQL (Kostrikov et al., 2022), TD3+BC (Fujimoto & Gu, 2021) and its improved version OAP (Yang et al., 2023), SRPO (Chen et al., 2024a), DQL (Wang et al., 2023), and DTQL (Chen et al., 2024c). We do not compare against additional value regularization methods (Huang et al., 2024), sequence modeling methods (Chen et al., 2021; Zhuang et al., 2024), or model-based methods (Yu et al., 2021), as they fall under fundamentally different methodological categories from ours. We only adjust the penalty coefficients $\beta_a$, $\beta_c$, and the bandwidth $\sigma$ in the Laplacian kernel for MMD calculation. Detailed hyperparameter configurations are provided in the Appendix.

### 4.2 RESULTS ON D4RL BENCHMARK

**Results on MuJoCo** The results of ReBEAR on MuJoCo datasets are summarized in Table 1. Overall, our method achieves superior performance compared to all competitors in this domain. Specif-

Table 2: Performance comparison on AntMaze '-v2' and Adroit '-v0' tasks. For AntMaze tasks, the results represent the mean and standard error of normalized scores over 5 random seeds (each evaluated with 100 trajectories), as these tasks generally exhibit high variance in performance. For Adroit tasks, the results are based on 5 random seeds (each evaluated with 10 trajectories). The abbreviations are: u = umaze, ud = umaze-diverse, mp = medium-play, md = medium-diverse, lp = large-play, ld = large-diverse, c = cloned and h = human.

| Task Name | CQL | IQL | TD3+BC | DQL | DTQL | BEAR | ReBEAR(Ours) |
|---|---|---|---|---|---|---|---|
| antmaze-u | 74.0 | 87.5 | 78.6 | 93.4 | 92.6 | 73.0 | $\mathbf{93.6} \pm 2.4$ |
| antmaze-ud | 84.0 | 62.2 | 71.4 | 66.2 | 74.4 | 61.0 | $\mathbf{90.6} \pm 3.2$ |
| antmaze-mp | 61.2 | 71.2 | 0.0 | 76.6 | 76.0 | 0.0 | $\mathbf{89.6} \pm 2.2$ |
| antmaze-md | 53.7 | 70.0 | 0.2 | 78.6 | 80.6 | 8.0 | $\mathbf{87.6} \pm 3.8$ |
| antmaze-lp | 15.8 | 39.6 | 0.0 | 46.4 | 59.2 | 0.0 | $\mathbf{73.0} \pm 7.3$ |
| antmaze-ld | 14.9 | 47.5 | 0.0 | 56.6 | 62.0 | 0.0 | $\mathbf{70.8} \pm 2.8$ |
| Total score | 303.6 | 378.0 | 150.2 | 417.8 | 444.8 | 142.0 | **505.2** |

| Task Name | BCQ | CQL | IQL | DQL | DTQL | BEAR | ReBEAR(Ours) |
|---|---|---|---|---|---|---|---|
| pen-c | 68.9 | 35.2 | 71.5 | 72.8 | 64.1 | -1.0 | $\mathbf{87.2} \pm 9.4$ |
| pen-h | 44.0 | 27.2 | 37.3 | 57.3 | 81.3 | 26.5 | $\mathbf{92.1} \pm 5.4$ |
| Total score | 112.9 | 62.4 | 108.8 | 130.1 | 145.4 | 25.5 | **179.3** |

ically, ReBEAR outperforms the state-of-the-art MSE loss-based constraint methods, TD3+BC and OAP, in 7 out of 9 tasks. While the diffusion-based method DTQL demonstrates strong competitive performance, ReBEAR still achieves a 4.7% improvement. It also surpasses the leading score regularized method SRPO in 8 out of 9 tasks. Moreover, compared to the previous MMD-based method BEAR, our approach delivers a substantial **103.2%** performance gain.

**Results on Complicated Tasks** The results of ReBEAR on AntMaze and Adroit datasets are summarized in Table 2. Overall, our method consistently achieves state-of-the-art performance across all tasks. Specifically, ReBEAR surpasses the previous best method, DTQL, by 13.6% on AntMaze and 23.3% on Adroit, demonstrating its effectiveness in handling more complex tasks. AntMaze particularly showcases ReBEAR's strong trajectory stitching capability, while Adroit demonstrates its ability to accurately capture narrow data distributions. Furthermore, compared to the previous MMD-based method, BEAR, our method achieves impressive improvements of **255.7%** on AntMaze and **603.1%** on Adroit. This clearly demonstrates the effectiveness of our designs.

## 4.3 SENSITIVE ANALYSES

We conduct detailed parameter analyses to examine the impact of the actor penalty coefficient $\beta_a$ and the target Q-value penalty coefficient $\beta_c$. Additionally, we explore the effect of the bandwidth $\sigma$ in the Laplacian kernel for MMD computation. Each experiment is performed using three different random seeds for each parameter setting.

### 4.3.1 POLICY CONSTRAINT COEFFICIENT

The policy constraint penalty coefficient $\beta_a$ regulates the extent to which the learned policy deviates from the behavior policy under MMD. To analyze its impact, we vary $\beta_a$ around its optimal value, as illustrated in Fig. 4. Overall, the results indicate that performance remains stable across a broad range of $\beta_a$ settings. Nevertheless, a more fine-grained analysis reveals task-specific trends. For medium-level tasks, increasing $\beta_a$ leads to a clear performance decline compared to other task levels. This is because a larger $\beta_a$ constrains the policy closer to the dataset, which contains more suboptimal behaviors. In contrast, for medium-expert tasks, a larger $\beta_a$ is generally preferable, as the dataset includes high-quality decisions. Meanwhile, medium-replay tasks exhibit the highest stability, benefiting from the diversity and richness of the data.

### 4.3.2 TARGET VALUE PENALTY COEFFICIENT

To assess the impact of the value penalty coefficient $\beta_c$, we conduct sensitivity analyses by varying $\beta_c$ around its optimal value. To better understand its influence, we use score curves to illustrate the performance across different settings, as illustrated in Fig. 5. The results reveal task-specific variations in performance. For the halfcheetah tasks, the performance remains consistently stable as

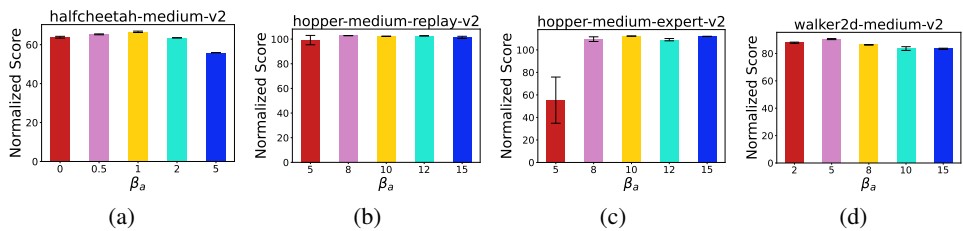

(a)  (b)  (c)  (d)

Figure 4: Performances of ReBEAR under different values of $\beta_a$.

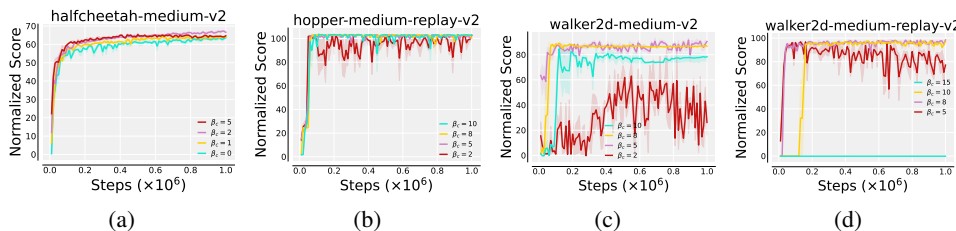

(a)  (b)  (c)  (d)

Figure 5: Performances of ReBEAR under different values of $\beta_c$.

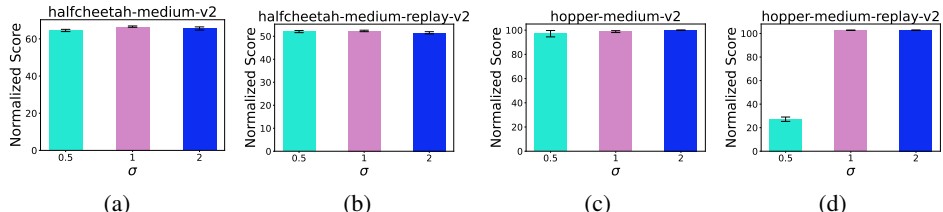

(a)  (b)  (c)  (d)

Figure 6: Performances of ReBEAR under different values of $\sigma$.

$\beta_c$ changes. In the hopper tasks, ReBEAR also converges to stable performance when $\beta_c$ is adjusted around its optimal value. However, for walker2d-medium-replay task (Fig. 5(d)), the largest value of $\beta_c$ lead to potential policy failure, indicating that an overly strong penalty may introduce value bias and cause the policy to degrade. Conversely, in the walker2d-medium task (Fig. 5(c)), the smallest value of $\beta_c$ causes a significant performance drop, demonstrating that slightly penalty fails to control overoptimism. Nonetheless, the performance for $\beta_c$ values closest to the optimal value remains stable. These observations align with the ablation studies (Section 4.4), where tasks experiencing such instability cannot perform well when relying solely on the actor penalty.

### 4.3.3 BANDWIDTH PARAMETER

To evaluate the impact of the bandwidth parameter $\sigma$ in the Laplacian kernel, we conduct a sensitivity analysis by varying $\sigma$ across $\{0.5, 1, 2\}$ on the MuJoCo tasks. As shown in Fig. 6, ReBEAR demonstrates stable performance across all tasks when $\sigma = 1$, making it a generally reliable choice for MuJoCo. In addition, $\sigma = 2$ can achieve similar stable and competitive performance. However, caution is needed with smaller $\sigma$ values, as they cause MMD to focus on fine-grained differences between points, potentially leading to increased sensitivity, as observed in Fig. 6(d).

### 4.4 ABLATION STUDIES ON TARGET VALUE PENALTY

To evaluate the contribution of the target Q-value penalty term (as defined in Eq. (7)), we conduct an ablation study. Specifically, we assess the performance of ReBEAR without the value penalty, focusing solely on the actor penalty. All experiments are conducted across three different random seeds, with the results presented in Fig. 7. As shown in Fig. 7(a) and (b), the performance without the value penalty remains comparable to ReBEAR's full implementation. This confirms that, as previously mentioned, MMD, when coupled with an accurate diffusion behavior model, can achieve high performance in most tasks. However, for the walker2d-medium-replay (Fig. 7(c)) and walker2d-medium (Fig. 7(d)) tasks, the normalized score curves reveal a notable performance degradation.

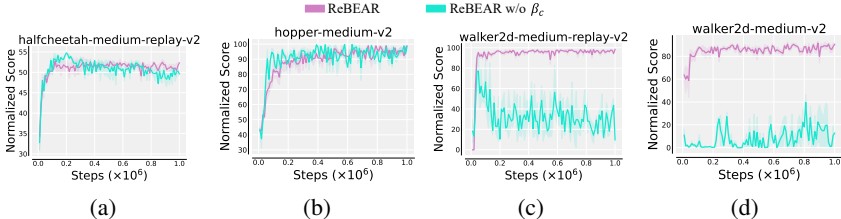

Figure 7: Performance comparison of the ReBEAR algorithm with and without the value penalty.

This demonstrates that, without penalizing overestimated values, the policy may exploit the support of suboptimal actions with inflated values. These results clearly highlight the effectiveness of the target Q-value penalty in mitigating such issues.

# 5 RELATED WORK

## 5.1 POLICY CONSTRAINT METHODS

Previous policy regularization methods employ various distribution constraints to align the learned policy with the behavior policy. BCQ (Fujimoto et al., 2019) constructs the policy by adding perturbations to a separately trained Conditional-VAE behavior-cloning model. BEAR (Kumar et al., 2019) introduces a weighted behavior-cloning loss into the policy improvement step by minimizing the MMD. BRAC (Wu et al., 2019) primarily uses KL divergence to constrain the policy, while TD3+BC (Fujimoto & Gu, 2021) adopts a similar approach to BEAR through MSE loss. DOGE (Li et al., 2023) and OAP (Yang et al., 2023) enhance TD3+BC by incorporating weighted MSE loss, while iTRPO (Zhang & Tan, 2024) improves BRAC by introducing an additional KL divergence term with the previous policy. Among these, BEAR is the most closely related to our work, as both employ MMD-based constraints. However, we empirically observe that direct cloning fails to match the support of the behavior policy, contrary to BEAR's argument. Our findings reveal that the MMD constraint exhibits strong mode-seeking capabilities when paired with accurate behavior modeling and value guidance. Building on this insight, we utilize a diffusion model for behavior policy representation and introduce a penalty on target Q-values, improving upon BEAR's framework.

## 5.2 DIFFUSION MODEL IN OFFLINE RL

Previous works have explored the use of diffusion models for policy modeling (DQL (Wang et al., 2023)) and trajectory generation (Diffuser (Janner et al., 2022), PlanCP (Sun et al., 2023)), respectively. However, the increased training and inference times associated with the denoising process have motivated subsequent studies to accelerate diffusion models in offline RL. For instance, EDP (Kang et al., 2023) introduces an approximate diffusion sampling scheme to reduce the number of required steps. IDQL (Hansen-Estruch et al., 2023) simplifies the training process by training a behavior cloning policy without the need for denoising sampling. Similarly, DTQL (Chen et al., 2024c) employs a diffusion model to represent the behavior policy and learns the optimal policy using classical offline RL methods. The most relevant works to ours are DQL and DTQL, as both adopt policy regularization strategies. However, our method differs by leveraging an MMD-based constraint, whereas DQL and DTQL rely on denoising loss for policy regularization.

# 6 CONCLUSION

In this work, we reveal crucial insights about MMD-based methods that were previously misunderstood in the offline RL community: (1) The lack of mode-seeking capability of MMD-based method may merely stems from previous distortion of behavior modeling. (2) Under diffusion behavior policy, MMD constraint demonstrates robust mode-seeking capability by leveraging Q-value guidance. By addressing the limitations associated with these findings, our method ReBEAR significantly outperforms the prior MMD-based method across all test domains. Furthermore, empirical evaluations demonstrate ReBEAR's state-of-the-art performance compared to the leading policy constraint methods. We hope that our work will inspire further exploration of MMD-based policy constraints and provide valuable insights into distilling knowledge from diffusion models using MMD.

## REPRODUCIBILITY STATEMENT

All technical details of our method are provided in the main text, with the complete set of hyperparameters included in the Appendix. We believe our results can be easily reproduced even from scratch. To further support reproducibility, we release the implementation of the 2D bandit example in an anonymous repository: `https://anonymous.4open.science/r/ReBEAR-0B73/`. The full codebase will be made publicly available upon the acceptance of this paper.

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

## A  MORE EXPERIMENTS ON THE 2D BANDIT EXAMPLE

### A.1  TWO MODES WITH THE SAME REWARD

We investigate an interesting scenario where two modes share the highest reward, testing whether ReBEAR captures one or both modes. Two settings are evaluated: (1) "left-up and right-up" actions with equal rewards (Fig. 8(a)), and (2) "left-up and right-down" actions with equal rewards (Fig. 8(b)). In the "right-angled edge" case (Fig. 8(a)), ReBEAR captures the segment between the two modes. For the "diagonal" case (left-up, right-down), only one mode is captured. This success in the diagonal case may stem from perturbation of the inter-mode connection by low-reward samples of other two modes, whereas the right-angled edge case remains unaffected by lower-reward actions, resulting in mode merging.

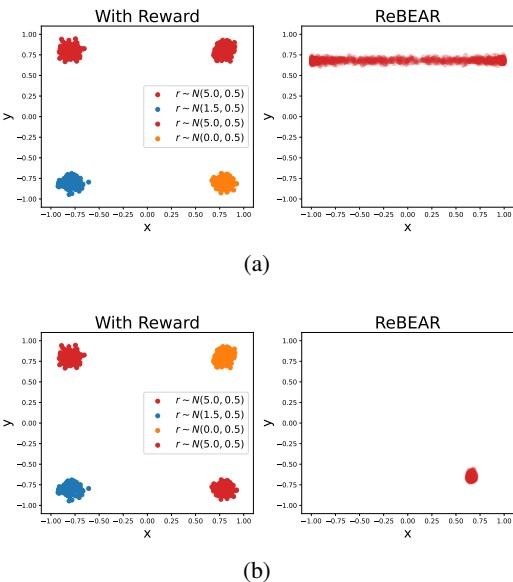

Figure 8: Mode-seeking experiments in the case of two modes with the same reward.

### A.2  MODE-SEEKING WITH DIFFERENT Q-VALUE WEIGHTS

As demonstrated in Fig. 9, ReBEAR consistently maintains strong mode-seeking capability across varying Q-value weights, while BEAR fails to converge to the optimal mode even at extreme weights (e.g., weight=50). In addition, ReBEAR's policy distribution shows desirable asymptotic behavior as Q-weighting increases: (1) The mean of learned policy converges to the ground-truth optimal value, and (2) the variance monotonically decreases, demonstrating stable policy refinement under increasing value guidance.

## B  THEORETICAL ANALYSIS OF MODE-SEEKING PROPERTY

In the context of the bandit example, the behavior policy $\pi_\beta$ is a Gaussian mixture distribution composed of $M$ Gaussian policies: $\pi_{\beta_j} \sim \mathcal{N}(a \in \mathbb{R}^D \mid \mu_j, \sigma_j^2 I)$, $j = 1, \cdots, M$. Assume each action sampled from the $j$-th mode policy $\pi_{\beta_j}$ yields a reward $r_j$, with rewards ordered as $r_1 > r_2 > \cdots > r_M$. Consequently, the expected Q-value for any action $a$ is given by $q(a) = \sum_{j=1}^M \pi_{\beta_j}(a) r_j$. For a learned policy $\pi$, Q-value maximization corresponds to maximize $\int \sum_{j=1}^M \pi_{\beta_j}(a) r_j \mathrm{d}\pi$. Under the MMD constraint, this yields the constrained optimization problem:

$$\max_\pi \mathbb{E}_{a \sim \pi}[q(a)]$$
$$\text{s.t. MMD}(\pi, \widehat{\pi}_\beta^N) \leq \tau, \tag{13}$$

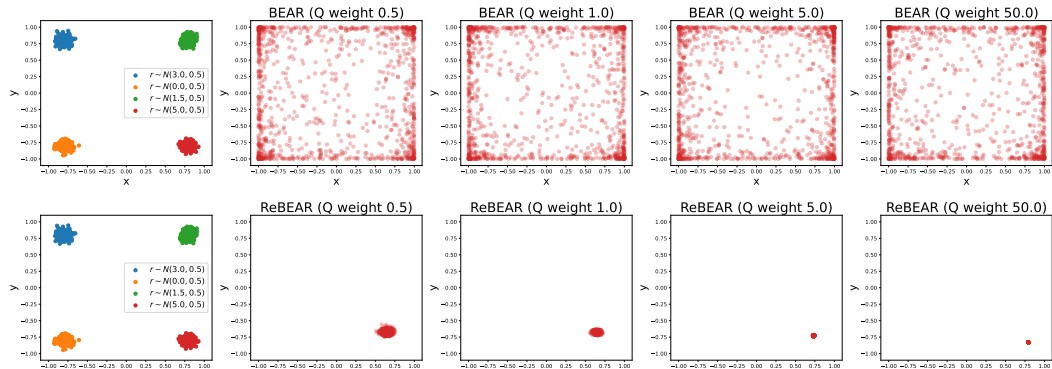

Figure 9: Mode-seeking behavior under different Q-value weighting configurations.

where $\tau$ controls the tightness of the policy constraint, and $\widehat{\pi}_\beta^N = \frac{1}{N} \sum_{i=1}^N \delta_{a_i}$ is the empirical behavior distribution, $\delta_{a_i}$ denotes the Dirac delta function for action samples $a_i$, $i = 1, \cdots, N$, and $N$ is the sample size. When the optimal value $q^*$ of (13) is sufficiently large (as visualized in Fig. 3), the learned policy $\pi$ exhibits mode-seeking behavior: its mean $\mu_\pi$ converges toward $\mu_1$.

Solving the constrained optimization problem (13) exactly within the MMD ball is theoretically challenging. To make progress, we consider a tractable subspace of candidate distributions of the form: $\pi = \sum_{i=1}^N \omega_i \delta_{a_i}, \omega \in \Delta^{N-1}$, where $\Delta^{N-1}$ denotes the $(N-1)$-dimensional probability simplex. This restriction offers several advantages: (1) The diffusion-based behavior policy naturally aligns with this subspace, as evidenced by both empirical results and theoretical analyses (Cheng et al., 2024) demonstrating its superior support matching properties. (2) The finite-dimensional parameterization transforms the original infinite-dimensional problem into a more manageable constrained optimization over the simplex. Therefore, a relaxed version of the original optimization problem is obtained

$$\max_\omega \sum_{i=1}^N \omega_i q(a_i)$$

$$\text{s.t. MMD}\left(\sum_{i=1}^N \omega_i \delta_{a_i}, \widehat{\pi}_\beta^N\right) \leq \tau$$

$$\sum_{i=1}^N \omega_i = 1$$

$$\omega_i \geq 0, \ \forall \, i = 1, \ldots, N.$$

(14)

The optimization problem in (13) searches over a strictly larger feasible set than (14), which immediately implies the optimal value of (13) is greater than the one of (14).

Through direct application of the MMD definition and its properties, we can rigorously derive that

$$\text{MMD}\left(\sum_{i=1}^N \omega_i \delta_{a_i}, \widehat{\pi}_\beta^N\right)^2 = \left\|\sum_{i=1}^N \omega_i k(a_i, \cdot) - \frac{1}{N} \sum_{i=1}^N k(a_i, \cdot)\right\|_{\mathcal{H}}^2$$

$$= \left\|\sum_{i=1}^N (\omega_i - \frac{1}{N}) k(a_i, \cdot)\right\|_{\mathcal{H}}^2,$$

(15)

where $k(\cdot, \cdot)$ denotes the kernel function defined in the MMD metric. Building upon the theoretical framework for distributionally robust optimization with MMD constraints established in Lemma 5.1 of (Staib & Jegelka, 2019), we now present the proof of our main theorem. For clarity, we first restate Theorem 3.1 as follows.

**Theorem B.1.** *Let $\vec{q}$ be the vector with $i$-th element $q(a_i)$ and kernel matrix $K$ with its element $K_{ij} = k(a_i, a_j)$ used in the MMD constraint. When $\tau$ satisfies the following conditions:*

$$\tau \geq \frac{q_0 - (\mathbf{1}^{\mathrm{T}}\vec{q})/N}{\sqrt{\vec{q}^{\mathrm{T}}K^{-1}\vec{q} - \frac{(\vec{q}^{\mathrm{T}}K^{-1}\mathbf{1})^2}{\mathbf{1}^{\mathrm{T}}K^{-1}\mathbf{1}}}} \text{ and } \tau \cdot K^{-1}\left(\frac{\mathbf{1}^{\mathrm{T}}K^{-1}\vec{q}}{\mathbf{1}^{\mathrm{T}}K^{-1}\mathbf{1}}\mathbf{1} - \vec{q}\right) \preceq \frac{\sqrt{\vec{q}^{\mathrm{T}}K^{-1}\vec{q} - \frac{(\mathbf{1}^{\mathrm{T}}K^{-1}\vec{q})^2}{\mathbf{1}^{\mathrm{T}}K^{-1}\mathbf{1}}}}{N}\mathbf{1}, \tag{16}$$

*where $\mathbf{1}$ is a vector with all elements are one and $\preceq$ indicates element-wise inequality, we can indeed conclude that $q^* \geq q_0$.*

*Proof.* For analytical convenience, we express the right-hand side of Eq. (15) in quadratic form: $(\omega - \frac{1}{N}\mathbf{1})^{\mathrm{T}}K(\omega - \frac{1}{N}\mathbf{1})$. This representation reveals that Problem (14) constitutes a linear program with quadratic constraints. By relaxing the non-negativity constraints $\omega_i \geq 0$, we obtain the simplified formulation:

$$\max_{\omega} \vec{q}^{\mathrm{T}}\omega$$
$$\text{s.t.} \left(\omega - \frac{1}{N}\mathbf{1}\right)^{\mathrm{T}} K \left(\omega - \frac{1}{N}\mathbf{1}\right) \leq \tau^2 \tag{17}$$
$$\mathbf{1}^{\mathrm{T}}\omega = 1$$

We will later demonstrate that this relaxation operation leaves the optimal value unchanged when the second condition on $\tau$ is satisfied. To address problem (17), we consider solving an equivalent but more compact formulation:

$$\max_{u} \vec{q}^{\mathrm{T}}u$$
$$\text{s.t. } u^{\mathrm{T}}Ku \leq \tau^2 \tag{18}$$
$$\mathbf{1}^{\mathrm{T}}u = 0$$

This equivalence can be obtained by substituting $u = \omega - \frac{1}{N}\mathbf{1}$ into (18). Notably, a subtle difference remains between these two optimization problems: The optimal value of problem (17) equals the sum of $\frac{1}{N}\mathbf{1}^{\mathrm{T}}\vec{q}$ and the optimal value of problem (18). From an optimization perspective, we can transform the constrained problem into an unconstrained formulation through the Lagrangian function:

$$\max_{u} \min_{\eta \geq 0, \lambda} \left\{\vec{q}^{\mathrm{T}}u - \eta(u^{\mathrm{T}}Ku - \tau^2) - \lambda\mathbf{1}^{\mathrm{T}}u\right\}, \tag{19}$$

where $\eta, \lambda$ are Lagrangian multiplier. It is obvious that the feasible region contains the point $u = 0$ as a relative interior point. According to Slater's condition, strong duality holds. The optimal value is equal to:

$$\min_{\eta \geq 0, \lambda} \left\{\eta\tau^2 + \max_{u} \left[\vec{q}^{\mathrm{T}}u - \eta u^{\mathrm{T}}Ku - \lambda\mathbf{1}^{\mathrm{T}}u\right]\right\}$$
$$= \min_{\eta \geq 0, \lambda} \left\{\eta\tau^2 + \max_{u} \left[-\eta u^{\mathrm{T}}Ku + (\vec{q} - \lambda\mathbf{1})^{\mathrm{T}}u\right]\right\}. \tag{20}$$

Under standard kernel choices such as the Gaussian or Laplacian kernels, the MMD kernel matrix $K$ is guaranteed to be positive definite. This property enables us to analytically solve the inner quadratic optimization problem $\max_{u} \left[-\eta u^{\mathrm{T}}Ku + (\vec{q} - \lambda\mathbf{1})^{\mathrm{T}}u\right]$. Accordingly, we find that the optimal solution $u^*$ satisfies:

$$u^* = \frac{1}{2\eta}K^{-1}(\vec{q} - \lambda\mathbf{1}), \tag{21}$$

and the corresponding optimal value is

$$\frac{1}{4\eta}(\vec{q} - \lambda\mathbf{1})^{\mathrm{T}}K^{-1}(\vec{q} - \lambda\mathbf{1}). \tag{22}$$

Hence, the overall problem (20) is

$$\min_{\eta \geq 0, \lambda} \left\{\eta\tau^2 + \frac{1}{4\eta}(\vec{q} - \lambda\mathbf{1})^{\mathrm{T}}K^{-1}(\vec{q} - \lambda\mathbf{1})\right\}. \tag{23}$$

The optimal dual variable $\lambda^*$ admits the closed-form solution: $\lambda^* = (\mathbf{1}^\mathrm{T} K^{-1} \vec{q})/(\mathbf{1}^\mathrm{T} K^{-1} \mathbf{1})$. This solution emerges from the stationarity condition when solving the dual problem. Regarding the $\eta$-dependent terms in the objective function, we note that both coefficients in the objective function are positive. This structure permits an elegant solution through the arithmetic mean-geometric mean inequality, which gives minimal value:

$$2\sqrt{\eta \tau^2 \cdot \frac{1}{4\eta}(\vec{q} - \lambda^* \mathbf{1})^\mathrm{T} K^{-1}(\vec{q} - \lambda^* \mathbf{1})}$$
$$= \tau \sqrt{(\vec{q} - \lambda^* \mathbf{1})^\mathrm{T} K^{-1}(\vec{q} - \lambda^* \mathbf{1})}. \tag{24}$$

The equality condition yields the optimal solution when

$$\eta^* \tau^2 = \frac{1}{4\eta^*}(\vec{q} - \lambda^* \mathbf{1})^\mathrm{T} K^{-1}(\vec{q} - \lambda^* \mathbf{1})$$
$$\implies \frac{1}{2\eta^*} = \frac{\tau}{\sqrt{(\vec{q} - \lambda^* \mathbf{1})^\mathrm{T} K^{-1}(\vec{q} - \lambda^* \mathbf{1})}}, \tag{25}$$

Now we show that $\omega^* \succeq \mathbf{0}$. We can easily compute that

$$\omega^* = u^* + \frac{1}{N}\mathbf{1}$$
$$= \frac{1}{2\eta^*} K^{-1}(\vec{q} - \lambda^* \mathbf{1}) + \frac{1}{N}\mathbf{1}$$
$$= \frac{1}{2\eta^*} K^{-1}(\vec{q} - \frac{\mathbf{1}^\mathrm{T} K^{-1} \vec{q}}{\mathbf{1}^\mathrm{T} K^{-1} \mathbf{1}}\mathbf{1}) + \frac{1}{N}\mathbf{1}$$
$$= \frac{\tau}{\sqrt{(\vec{q} - \lambda^* \mathbf{1})^\mathrm{T} K^{-1}(\vec{q} - \lambda^* \mathbf{1})}} K^{-1}(\vec{q} - \frac{\mathbf{1}^\mathrm{T} K^{-1} \vec{q}}{\mathbf{1}^\mathrm{T} K^{-1} \mathbf{1}}\mathbf{1}) + \frac{1}{N}\mathbf{1}$$
$$= \frac{1}{\sqrt{(\vec{q} - \lambda^* \mathbf{1})^\mathrm{T} K^{-1}(\vec{q} - \lambda^* \mathbf{1})}} \left[ -\tau K^{-1}(\frac{\mathbf{1}^\mathrm{T} K^{-1} \vec{q}}{\mathbf{1}^\mathrm{T} K^{-1} \mathbf{1}}\mathbf{1} - \vec{q}) + \frac{\sqrt{(\vec{q} - \lambda^* \mathbf{1})^\mathrm{T} K^{-1}(\vec{q} - \lambda^* \mathbf{1})}}{N}\mathbf{1} \right]$$
$$\succeq \mathbf{0} \tag{26}$$

The final inequality holds by the second condition on $\tau$. Returning to our proof, we substitute $\lambda^*$ into Eq. (24) yields the overall optimal value

$$\tau \sqrt{(\vec{q} - \lambda^* \mathbf{1})^\mathrm{T} K^{-1}(\vec{q} - \lambda^* \mathbf{1})} = \tau \sqrt{\vec{q}^\mathrm{T} K^{-1} \vec{q} - 2\lambda^* \mathbf{1}^\mathrm{T} K^{-1} \vec{q} + (\lambda^*)^2 \mathbf{1}^\mathrm{T} K^{-1} \mathbf{1}}$$
$$= \tau \sqrt{\vec{q}^\mathrm{T} K^{-1} \vec{q} - \frac{(\mathbf{1}^\mathrm{T} K^{-1} \vec{q})^2}{\mathbf{1}^\mathrm{T} K^{-1} \mathbf{1}}}, \tag{27}$$

Therefore, the optimal value of problem (17) is

$$q_{\mathrm{lb}} = \frac{1}{N}\mathbf{1}^\mathrm{T} \vec{q} + \tau \sqrt{\vec{q}^\mathrm{T} K^{-1} \vec{q} - \frac{(\mathbf{1}^\mathrm{T} K^{-1} \vec{q})^2}{\mathbf{1}^\mathrm{T} K^{-1} \mathbf{1}}} \tag{28}$$

Combining $\tau \geq q_0 - (\mathbf{1}^\mathrm{T} \vec{q})/N \Big/ \sqrt{\vec{q}^\mathrm{T} K^{-1} \vec{q} - \frac{(\vec{q}^\mathrm{T} K^{-1} \mathbf{1})^2}{\mathbf{1}^\mathrm{T} K^{-1} \mathbf{1}}}$ with the above equation, we can easily derive that $q_{\mathrm{lb}} \geq q_0$. As discussed earlier, the search space of problem (17) is constrained to a smaller subset compared to the original feasible region, we thus have $q^* \geq q_{\mathrm{lb}}$. Consequently, the inequality stated in the theorem holds. $\qquad \square$

We now provide a concrete example to demonstrate $q^* \geq q_0$, based on the results established in the theorem, and to verify that the required conditions on $\tau$ can be readily satisfied.

For simplicity, we validate the result in a one-dimensional space. Let $\pi_{\beta_1} \sim \mathcal{N}(1, 0.5^2)$ and $\pi_{\beta_2} \sim \mathcal{N}(-1, 0.5^2)$. The action samples are $a_1 = 1$, $a_2 = -1$ with corresponding rewards $r_1 = 5$, $r_2 = 0$, respectively. We aim to compute the lower bound

$$q_{\mathrm{lb}} = \frac{1}{N}\mathbf{1}^\mathrm{T} \vec{q} + \tau \sqrt{\vec{q}^\mathrm{T} K^{-1} \vec{q} - \frac{(\mathbf{1}^\mathrm{T} K^{-1} \vec{q})^2}{\mathbf{1}^\mathrm{T} K^{-1} \mathbf{1}}} \tag{29}$$

and

$$q_0 = \sum_{j=1}^{2} r_j \cdot \frac{1}{\sqrt{2\pi\sigma_j^2}} \exp\left(-(\mu_1 + de_0 - \mu_j)^2 / 2\sigma_j^2\right). \tag{30}$$

To compute the lower bound, we need to evaluate both the vector $\vec{q}$ and the kernel matrix $K$. First, it is easy to obtain that:

$$
\begin{aligned}
\pi_{\beta_1}(a_1) &= \frac{1}{\sqrt{2\pi}\sigma_1} \exp\left(-\frac{(a_1 - \mu_1)^2}{2\sigma_1^2}\right) = \frac{2}{\sqrt{2\pi}} \exp(0) \approx 0.797885, \\
\pi_{\beta_1}(a_2) &= \frac{1}{\sqrt{2\pi}\sigma_1} \exp\left(-\frac{(a_2 - \mu_1)^2}{2\sigma_1^2}\right) = \frac{2}{\sqrt{2\pi}} \exp(-8) \approx 0.000268, \\
\pi_{\beta_2}(a_1) &= \frac{1}{\sqrt{2\pi}\sigma_2} \exp\left(-\frac{(a_1 - \mu_2)^2}{2\sigma_2^2}\right) = \frac{2}{\sqrt{2\pi}} \exp(-8) \approx 0.000268, \\
\pi_{\beta_2}(a_2) &= \frac{1}{\sqrt{2\pi}\sigma_2} \exp\left(-\frac{(a_2 - \mu_2)^2}{2\sigma_2^2}\right) = \frac{2}{\sqrt{2\pi}} \exp(0) \approx 0.797885.
\end{aligned}
\tag{31}
$$

According to the definition of $q(a)$, we have

$$
\begin{aligned}
q(a_1) &\approx 5 \cdot 0.797885 + 0 \cdot 0.000268 = 3.989425, \\
q(a_2) &\approx 5 \cdot 0.000268 + 0 \cdot 0.797885 = 0.00134.
\end{aligned}
\tag{32}
$$

Therefore,

$$\vec{q} = \begin{bmatrix} 3.989425 \\ 0.00134 \end{bmatrix}. \tag{33}$$

For this example, we employ a Gaussian kernel with bandwidth $\sigma = 0.5$, which yields

$$K = \begin{bmatrix} \exp(0) & \exp(-8) \\ \exp(-8) & \exp(0) \end{bmatrix} \approx \begin{bmatrix} 1 & 0.000335 \\ 0.000335 & 1 \end{bmatrix}. \tag{34}$$

The inverse of the kernel matrix $K$ is then given by:

$$K^{-1} \approx \begin{bmatrix} 1 & -0.000335 \\ -0.000335 & 1 \end{bmatrix}. \tag{35}$$

We now present the results of the intermediate terms as follows:

$$K^{-1}\vec{q} \approx \begin{bmatrix} 1 \cdot 3.989425 - 0.000335 \cdot 0.00134 \\ -0.000335 \cdot 3.989425 + 1 \cdot 0.00134 \end{bmatrix} = \begin{bmatrix} 3.989425 \\ 0.000004 \end{bmatrix}, \tag{36}$$

$$
\begin{aligned}
\vec{q}^{\mathrm{T}} K^{-1} \vec{q} &\approx 3.989425 \cdot 3.989425 + 0.00134 \cdot 0.000004 \approx 15.915512, \\
\mathbf{1}^{\mathrm{T}} K^{-1} \vec{q} &\approx 3.9893995 + 0.0000001 = 3.989429, \\
\mathbf{1}^{\mathrm{T}} K^{-1} \mathbf{1} &\approx 1 + 1 - 2 \cdot 0.000335 = 1.99933.
\end{aligned}
\tag{37}
$$

Finally, the lower bound of the optimal value is obtained

$$
\begin{aligned}
\frac{1}{N}\mathbf{1}^{\mathrm{T}}\vec{q} + \tau \cdot \sqrt{\vec{q}^{\mathrm{T}}K^{-1}\vec{q} - \frac{(\mathbf{1}^{\mathrm{T}}K^{-1}\vec{q})^2}{\mathbf{1}^{\mathrm{T}}K^{-1}\mathbf{1}}} &\approx 1.995383 + \tau \cdot \sqrt{15.915512 - \frac{3.989429^2}{1.99933}} \\
&\approx 1.995383 + \tau \cdot 2.820474.
\end{aligned}
\tag{38}
$$

It should be noted that we must select an appropriate value of $\tau$ to ensure all elements of $\omega^* = u^* + \frac{1}{N}\mathbf{1}$ are non-negative. We then compute $\lambda^*$ and $1/2\eta^*$ as follows

$$
\begin{aligned}
\lambda^* &= \frac{\mathbf{1}^{\mathrm{T}}K^{-1}\vec{q}}{\mathbf{1}^{\mathrm{T}}K^{-1}\mathbf{1}} \approx \frac{3.989429}{1.99933} \approx 1.995381, \\
\frac{1}{2\eta^*} &= \frac{\tau}{\sqrt{(\vec{q} - \lambda^*\mathbf{1})^{\mathrm{T}}K^{-1}(\vec{q} - \lambda^*\mathbf{1})}} \approx \tau \cdot 0.354550.
\end{aligned}
\tag{39}
$$

Accordingly,

$$u^* \approx \tau \cdot 0.354550 \cdot \begin{bmatrix} 1.994712 \\ -1.994709 \end{bmatrix} \approx \tau \cdot \begin{bmatrix} 0.707225 \\ -0.707224 \end{bmatrix} \tag{40}$$

Evidently, $\tau = 0.5$ suffices to maintain positivity of all elements in $\omega^*$, yielding the lower bound $1.995383 + 0.5 \cdot 2.820474 = 3.40562$. In addition, one can readily derive $q_0 \approx 0.539910$ when $d = 1$ and $e_0 = -1$. The results clearly demonstrate that $q^*$ is much larger than $q_0$.

## C   EXPERIMENTAL DETAILS

**Environments** We conduct extensive experiments on MuJoCo, AntMaze and Adroit tasks from D4RL benchmark (Fu et al., 2020). For MuJoCo, we evaluate our method on halfcheetah, hopper, and walker2d using the medium-expert, medium-replay, and medium datasets. Medium-expert consists of both expert and suboptimal data, while medium dataset is collected from an unconverged policy; medium-replay dataset includes all samples in the replay buffer during training until medium level policy. For AntMaze, we test across all dataset variants spanning three sizes—umaze, medium, and large—each with 'play' and 'diverse' types. In Adroit-pen, we assess performance on the human and cloned datasets, which correspond to human demonstrations and imitation-based policies, respectively. Compared to MuJoCo, AntMaze presents a greater challenge due to its long-horizon tasks and sparse rewards, while Adroit is more demanding due to its narrow distribution.

**Hyperparameters** In the following, we mainly present our implementation and experimental details. The basic hyperparameters associated with network architectures and training parameters for all tasks are listed in Table 3. The network architectures and settings for diffusion behavior policy follow the Diffusion Q-learning setup (Wang et al., 2023) with learning rate $3 \times 10^{-4}$ and training steps $3 \times 10^{5}$ for all tasks. All other elaborate settings, such as different learning rates for different tasks, different training steps, using decay in learning rate, using target policy or entropy in prior methods (Chen et al., 2024c; Kumar et al., 2019; 2020), have been removed, as summarized in Table 4. We only adjust the penalty coefficients $\beta_a$, $\beta_c$, and the bandwidth $\sigma$ in the Laplacian kernel for MMD calculation.

Table 3: Hyperparameters of ReBEAR

| Hyperparameters | Value |
|---|---|
| Actor Architecture | input-256-256-256-output |
| Critic Architecture | input-256-256-256-1 |
| Optimizer | Adam (Kingma & Ba, 2014) |
| Batch size | 256 |
| Critic learning rate | $5 \times 10^{-4}$ |
| Actor learning rate | $3 \times 10^{-4}$ |
| Number of Critics | 2 |
| Number of samples in MMD | 10 |
| Training steps | $10^6$ |
| Discount factor $\gamma$ | 0.999 for AntMaze, 0.99 others |
| Target update rate $\tau$ | 0.005 |

Table 4: ReBEAR operates WITHOUT any elaborate configurations for ALL tasks.

| Fixed learning rate | Without learning decay | Without large batch | Without policy warmup |
|---|---|---|---|
| ✓ | ✓ | ✓ | ✓ |
| Fixed learning steps | Without entropy | Without ensemble Q | Without target policy |
| ✓ | ✓ | ✓ | ✓ |

### C.1   HYPERPARAMETER SETTINGS ON MUJOCO TASKS

In our experiments, we focused on tuning three key hyperparameters: the actor penalty coefficient $\beta_a$, the critic penalty coefficient $\beta_c$, and the kernel bandwidth $\sigma$. (1) For task-specific tuning of $\beta_a$: For medium-expert level tasks, we selected $\beta_a$ from $\{5, 10\}$ to encourage the learned policy to remain close to the behavior policy; For medium and medium-replay levels of halfcheetah, we tuned $\beta_a$ within $\{1, 5\}$ due to its training stability observed from other methods; For hopper and walker2d at medium and medium-replay levels, we evaluated $\beta_a$ values from $\{5, 8, 10\}$. (2) The value penalty coefficient $\beta_c$ was similarly tuned according to task characteristics: halfcheetah tasks, being training

stable, used $\beta_c \in \{0, 2\}$; Medium and medium-replay tasks for hopper and walker2d employed $\beta_c \in \{1, 5, 8\}$; Medium-expert tasks for these environments utilized $\beta_c \in \{5, 8, 10\}$ to control potential value overestimation in bootstrap. (3) For the bandwidth parameter $\sigma$, we initially set $\sigma = 1$ during preliminary tuning, which provided strong performance. Through subsequent sensitivity analysis on $\sigma$, we identified more optimal settings. The final hyperparameter configurations for all MuJoCo tasks are provided in Table 5.

Table 5: Main hyperparameters of ReBEAR on MuJoCo

| Task | $\beta_a$ | $\beta_c$ | $\sigma$ |
|---|---|---|---|
| halfcheetah-medium-v2 | 1 | 2 | 1 |
| hopper-medium-v2 | 10 | 1 | 1 |
| walker2d-medium-v2 | 5 | 5 | 1 |
| halfcheetah-medium-replay-v2 | 1 | 2 | 1 |
| hopper-medium-replay-v2 | 8 | 5 | 2 |
| walker2d-medium-replay-v2 | 8 | 8 | 2 |
| halfcheetah-medium-expert-v2 | 10 | 0 | 1 |
| hopper-medium-expert-v2 | 10 | 10 | 0.5 |
| walker2d-medium-expert-v2 | 5 | 5 | 1 |

### C.2 HYPERPARAMETER SETTINGS ON ANTMAZE TASKS

In the AntMaze tasks, we tuned the parameter $\beta_a$ within the range of $\{0.5, 0.8, 1\}$ since their Q-values are smaller than those of MuJoCo tasks. The $\beta_c$ is adjusted within the set $\{0, 0.05\}$. The parameter $\sigma$ utilized in MMD is tuned within the set $\{1, 2, 5\}$. The optimal hyperparameter settings for AntMaze tasks are listed in Table 6. We also scale the reward by multiplying it by 100, consistent with other methods. This adjustment is necessary because episodes in AntMaze can extend up to 1,000 steps, with sparse rewards that are only obtained at the end of an episode. Scaling the reward facilitates reward signal propagation along the trajectories, improving the learning process.

Table 6: Main hyperparameters of ReBEAR on AntMaze

| Task | $\beta_a$ | $\beta_c$ | $\sigma$ |
|---|---|---|---|
| antmaze-umaze-v2 | 1 | 0 | 2 |
| antmaze-umaze-diverse-v2 | 0.5 | 0 | 2 |
| antmaze-medium-play-v2 | 1 | 0.05 | 1 |
| antmaze-medium-diverse-v2 | 0.8 | 0 | 5 |
| antmaze-large-play-v2 | 0.5 | 0 | 1 |
| antmaze-large-diverse-v2 | 0.5 | 0 | 2 |

### C.3 HYPERPARAMETER SETTINGS ON ADROIT TASKS

In the Adroit tasks, we tuned the parameter $\beta_a$ within the set $\{200, 500, 800, 1000, 1500\}$, as their Q-values are very large. We doesn't fine-tune the $\beta_c$ and set it to 0 directly. The parameter $\sigma$ utilized in MMD is tuned within the set $\{0.5, 1, 2\}$. The optimal hyperparameter settings for Adroit tasks are listed in Table 7.

Table 7: Main hyperparameters of ReBEAR on Android

| Task | $\beta_a$ | $\beta_c$ | $\sigma$ |
|---|---|---|---|
| pen-cloned-v0 | 500 | 0 | 2 |
| pen-human-v0 | 1500 | 0 | 2 |
| pen-expert-v0 | 800 | 0 | 1 |

## C.4 COMPUTING INFRASTRUCTURE

The computing devices and platforms are listed as follows:

- OS: Ubuntu 18.04.6 LTS

- GPU: NVIDIA Tesla V100

- GPU Memory: 32 G

- Python 3.8.13

- PyTorch 1.12.1

# D PSEUDOCODE OF REBEAR

---
**Algorithm 1** ReBEAR
---
**Require:** Dataset $\mathcal{D}$, number of steps $N$, Number of steps for behavior modeling $N_b$, discount factor $\gamma$, target network update rate $\tau$, actor penalty coefficient $\beta_a$, target Q-value penalty $\beta_c$ and bandwidth parameter $\sigma$.
1: Initialize critic networks $Q_{\theta_1}, Q_{\theta_2}$, actor network $\pi_\phi$, diffusion model $\xi_\omega$, target networks $Q_{\theta_1^-}$, $Q_{\theta_2^-}$ with $\theta_i^- \leftarrow \theta_i, i = \{1, 2\}$.
2: // Training the diffusion model
3: **for** step $n = 1$ to $N_b$ **do**
4:    Train the diffusion model $\xi_\omega$ for modeling the behavior policy by optimizing (6).
5: **end for**
6: // Actor-Critic
7: **for** step $n = 1$ to $N$ **do**
8:    Sample a mini-batch of transitions $\mathcal{B} = \{(s, a, r, s')\}$ from dataset $\mathcal{D}$.
9:    Compute target value as (7).
10:    Update parameters $\theta_i, i = 1, 2$ for each critic network via minimizing (8).
11:    Update actor $\phi$ via (9).
12:    Update the target networks: $\theta_i^- \leftarrow \tau\theta_i + (1 - \tau)\theta_i^-, i = 1, 2$.
13: **end for**
---

# E NORMALIZED SCORE CURVES

Figures 10, 11, and 12 present the normalized score curves during training, offering a comprehensive overview of performance across all tasks.

# F MORE EXPERIMENTAL RESULTS

## F.1 SENSITIVITY ANALYSES OF NUMBER OF SAMPLES

In the definition of MMD, $m$ and $n$ denote the sample sizes for the two distributions. For our experiments, we set $m = n = 10$ as the default configuration. To evaluate parameter robustness, we further conduct sensitivity analyses across three sample size configurations ($m = n = 5, 10, 15$) as detailed in Fig. 13. Empirical results confirm that model performance remains statistically stable across varying sample sizes.

## F.2 PERFORMANCE UNDER FIXED BANDWIDTH AND ENVIRONMENT-SPECIFIC VALUE PENALTY PARAMETERS

While pursuing concise hyperparameter configurations, our method still maintains strong performance. We fix the bandwidth parameter ($\sigma = 1$) across all MuJoCo tasks for simplicity. For the value penalty $\beta_c$, we adopt environment-specific defaults: 1 for halfcheetah, 2 for hopper, and 8 for walker2d tasks. The policy constraint coefficient $\beta_a$ retains its intuitive interpretation - typically

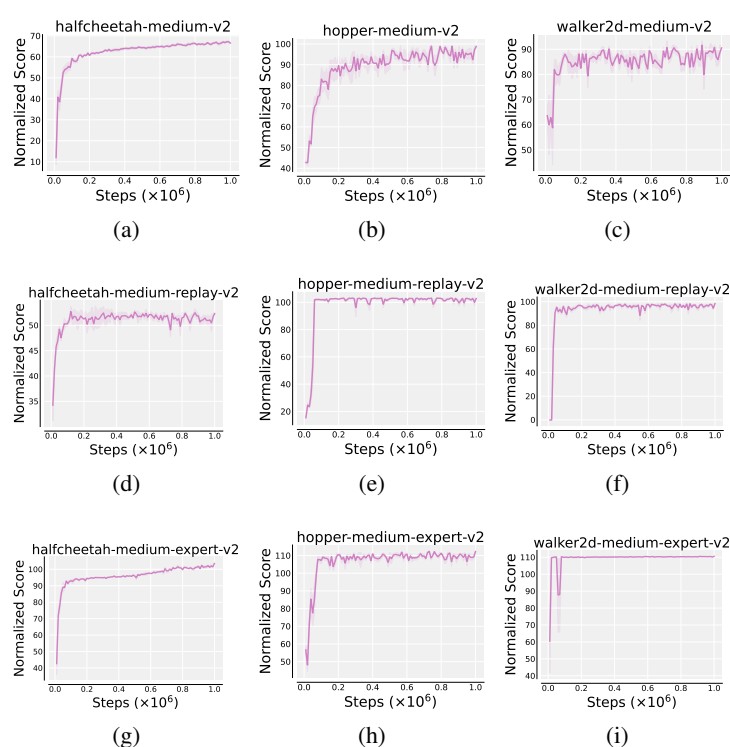

Figure 10: Normalized score curves of ReBEAR on MuJoCo. The results are based on 50 random rollouts (5 independently trained models, each evaluated with 10 trajectories). The Shaded area of each curve indicate its standard deviation.

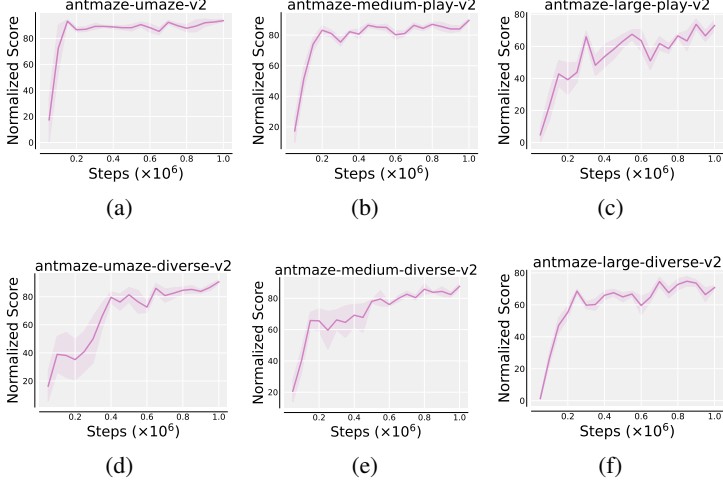

Figure 11: Normalized score curves of ReBEAR on AntMaze. The results are based on 500 random rollouts (5 independently trained models, each evaluated with 100 trajectories). The Shaded area of each curve indicate its standard deviation.

requiring stronger penalties when the dataset policy approaches expert quality - and thus we preserve the optimal settings from Table 5. Under these simplified settings, performance of our method across all MuJoCo tasks are demonstrated in Table 8.

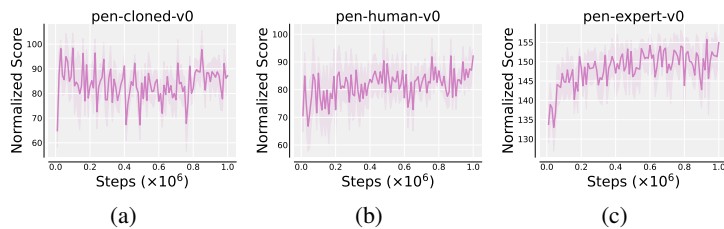

(a)            (b)            (c)

Figure 12: Normalized score curves of ReBEAR on Adroit. The results are based on 50 random rollouts (5 independently trained models, each evaluated with 10 trajectories). The Shaded area of each curve indicate its standard deviation.

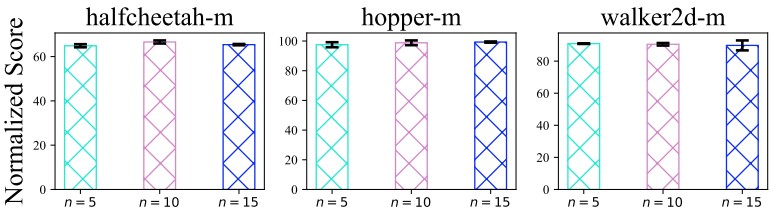

Figure 13: Performances of ReBEAR under different sample sizes

Table 8: Performance on MuJoCo under fixed bandwidth $\sigma$ and environment-specific value penalty $\beta_c$.

| Task Name | DQL | DTQL | BEAR | ReBEAR(Optimal) | ReBEAR(Fixed) |
|---|---|---|---|---|---|
| halfcheetah-m | 51.1 | 57.9 | 41.0 | **66.6** $\pm$ 0.7 | 64.0 $\pm$ 1.0 |
| hopper-m | 90.5 | **99.6** | 51.9 | 98.8 $\pm$ 1.6 | 95.3 $\pm$ 3.3 |
| walker2d-m | 87.0 | 89.4 | 80.9 | **90.5** $\pm$ 0.8 | 86.7 $\pm$ 0.2 |
| halfcheetah-mr | 47.8 | 50.9 | 29.7 | **52.3** $\pm$ 0.7 | 51.3 $\pm$ 0.4 |
| hopper-mr | 101.3 | 100.0 | 37.3 | **102.8** $\pm$ 0.2 | 102.3 $\pm$ 0.4 |
| walker2d-mr | 95.5 | 88.5 | 18.5 | **98.4** $\pm$ 1.3 | 97.4 $\pm$ 0.8 |
| halfcheetah-me | 96.8 | 92.7 | 38.9 | **103.6** $\pm$ 1.3 | 101.1 $\pm$ 0.2 |
| hopper-me | 111.1 | 109.3 | 17.7 | **112.3** $\pm$ 0.4 | 102.0 $\pm$ 7.2 |
| walker2d-me | 110.1 | 110.0 | 95.4 | **110.5** $\pm$ 0.2 | 110.4 $\pm$ 0.1 |
| Total score | 791.2 | 798.3 | 411.3 | **835.8** | 810.5 |

## F.3   PERFORMANCE OF REBEAR ON ANTMAZE-V0

Some prior works reported results on the AntMaze '-v0' datasets. To enable a more comprehensive comparison, we additionally evaluated ReBEAR on all tasks in the AntMaze-v0 domain, with results averaged over five random seeds. As shown in the Table 9, ReBEAR consistently achieves strong performance on the v0 datasets as well, further demonstrating the effectiveness of our method.

Table 9: Performance on AntMaze '-v0' domain.

| Task Name | IQL | DTQL | ReBEAR |
|---|---|---|---|
| antmaze-v0-u | 87.5 | 90.8 | **96.0**$\pm$1.5 |
| antmaze-v0-ud | 62.2 | 59.0 | **88.0**$\pm$5.8 |
| antmaze-v0-mp | 71.2 | 73.0 | **85.2**$\pm$4.4 |
| antmaze-v0-md | 70.0 | 65.2 | **82.0**$\pm$6.4 |
| antmaze-v0-lp | 39.6 | 38.8 | **58.8**$\pm$7.2 |
| antmaze-v0-ld | 47.5 | 33.8 | **70.2**$\pm$6.2 |

## F.4 MORE RESULTS OF SENSITIVITY ANALYSES

We provide additional sensitivity analyses of both $\beta_a$ and $\beta_c$ across a broader set of tasks; see Fig. 14 and Fig. 15.

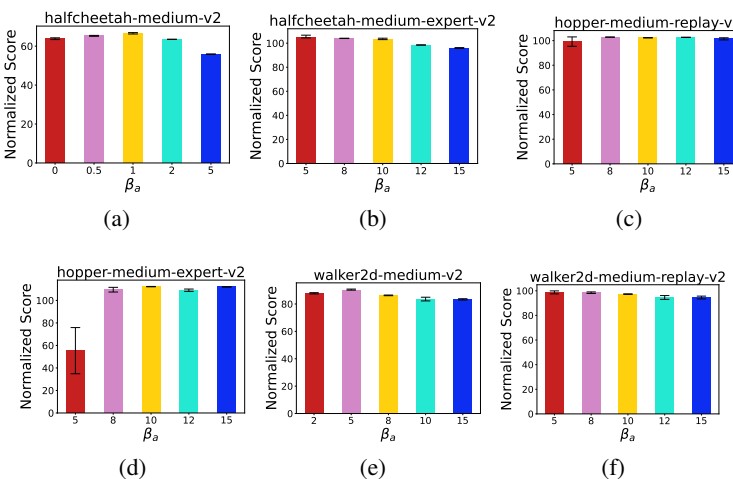

Figure 14: Performances of ReBEAR under different values of $\beta_a$.

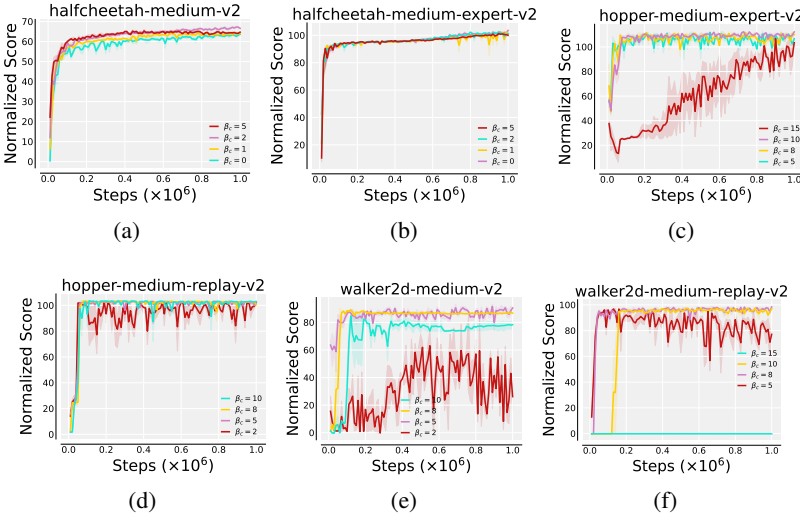

Figure 15: Performances of ReBEAR under different values of $\beta_c$.

## F.5 MORE RESULTS OF ABLATION STUDIES

We present additional ablation experiments on a wider range of tasks, as shown in Fig. 16.

## F.6 SENSITIVITY TO THE CHOICE OF KERNEL FUNCTION IN THE MMD CONSTRAINT

We consistently use the same Laplacian kernel across all 17 tasks and observe strong, stable performance, indicating broad adaptability. To assess sensitivity to kernel choice, we further conducted experiments with a Gaussian kernel on three tasks. The results, averaged over three random seeds, show performance comparable to that of the Laplacian kernel, as reported in the Table 10.

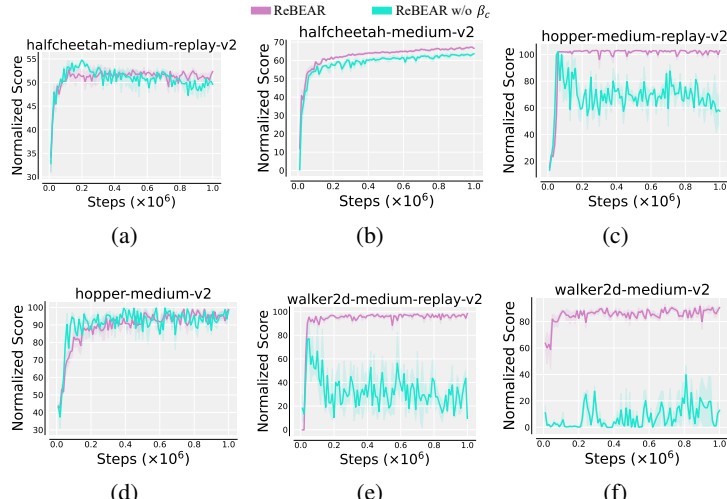

Figure 16: Performance comparison of the ReBEAR algorithm with and without the value penalty.

Table 10: Comparison of Laplacian and Gaussian kernels. Both kernels achieve comparable performance, demonstrating the robustness of ReBEAR to kernel choice.

| Task | Laplacian kernel | Gaussian kernel |
|---|---|---|
| halfcheetah-m | 66.6±0.7 | 65.6±0.8 |
| hopper-mr | 102.8±0.2 | 102.0±1.3 |
| walker2d-me | 110.5±0.2 | 111.3±0.1 |

## F.7 ADDITIONAL COMPARISONS WITH OTHER DIFFUSION-BASED METHODS

We have already compared ReBEAR with the most relevant diffusion-based methods, including DQL, DTQL, and SRPO. To ensure comprehensiveness, we further extend our comparison to other diffusion-based approaches, namely Diffuser (Janner et al., 2022), DD (Ajay et al., 2023), SfBC (Chen et al., 2023), IDQL (Hansen-Estruch et al., 2023), IQL+EDP, TD3+BC+EDP (Kang et al., 2023), QGPO (Lu et al., 2023), and BDM (Chen et al., 2024b). The results on the MuJoCo and AntMaze domains are presented in Tables 11 and 12, respectively.

Table 11: Comparison with other diffusion-based methods on MuJoCo domain.

| Task | Diffuser | DD | SfBC | IDQL | IQL+EDP | TD3+BC+EDP | QGPO | BDM | ReBEAR (ours) |
|---|---|---|---|---|---|---|---|---|---|
| halfcheetah-m | 44.2 | 49.1 | 45.9 | 51.0 | 48.1 | 52.1 | 54.1 | 57.0 | **66.6±0.7** |
| hopper-m | 58.5 | 79.3 | 57.1 | 65.4 | 63.1 | 81.9 | 98.0 | 98.4 | **98.8±1.6** |
| walker2d-m | 79.7 | 82.5 | 77.9 | 82.5 | 85.4 | 86.9 | 86.0 | 87.4 | **90.5±0.8** |
| halfcheetah-mr | 42.2 | 39.3 | 37.1 | 45.9 | 43.8 | 49.4 | 47.6 | 51.6 | **52.3±0.7** |
| hopper-mr | 96.8 | 100.0 | 86.2 | 92.1 | 99.1 | 101.0 | 96.9 | 92.7 | **102.8±0.2** |
| walker2d-mr | 61.2 | 75.0 | 65.1 | 85.1 | 84.0 | 94.9 | 84.4 | 89.2 | **98.4±1.3** |
| halfcheetah-me | 79.8 | 90.6 | 92.6 | 95.9 | 86.7 | 95.5 | 93.5 | 93.2 | **103.6±1.3** |
| hopper-me | 107.2 | 111.8 | 108.6 | 108.6 | 99.6 | 97.4 | 108.0 | 104.9 | **112.3±0.4** |
| walker2d-me | 108.4 | 108.8 | 109.8 | **112.7** | 109.0 | 110.2 | 110.7 | 111.1 | 110.5±0.2 |
| Total Score | 678.0 | 736.4 | 680.3 | 739.2 | 718.8 | 769.3 | 779.2 | 785.5 | **835.8** |

## F.8 COMPUTATION COST: TRAINING AND INFERENCE

As discussed in our limitations (in Section G), MMD calculation introduces computational overhead by requiring samples from the behavior diffusion policy. We measure this impact through comparative training times in Table 13, where $T_b$ represents behavior policy training time. BEAR incurs the highest training time in $T$ due to integrated CVAE encoder/decoder training within actor-critic updates. Both DTQL and ReBEAR require separate behavior diffusion policy training $T_b$. As shown

Table 12: Comparison with other diffusion-based methods on AntMaze domain.

| Task | SfBC | IDQL | IQL+EDP | TD3+BC+EDP | QGPO | ReBEAR (ours) |
|------|------|------|---------|-----------|------|---------------|
| antmaze-u | 92.0 | 94.0 | 94.2 | **96.6** | 96.4 | 93.6±1.4 |
| antmaze-ud | 85.3 | 80.2 | 79.0 | 69.5 | 74.4 | **90.6±3.2** |
| antmaze-mp | 81.3 | 84.5 | 81.8 | 0.0 | 83.6 | **89.6±2.2** |
| antmaze-md | 82.0 | 84.8 | 82.3 | 6.4 | 83.8 | **87.6±3.8** |
| antmaze-lp | 59.3 | 63.5 | 42.3 | 1.6 | 66.6 | **73.0±7.3** |
| antmaze-ld | 45.5 | 67.9 | 60.6 | 4.4 | 64.8 | **70.8±2.8** |
| Total Score | 445.4 | 474.9 | 440.2 | 178.5 | 469.6 | **505.2** |

in Table 13, while MMD computation in ReBEAR introduces additional computational cost, the impact remains manageable due to using only 5 denoising timesteps.

Table 13: Training time (seconds) per 100 steps for different algorithms on the hopper-medium-v2 task. All experiments are conducted with the same computing hardware and system.

| | BEAR | DQL | DTQL | ReBEAR |
|------|------|------|------|--------|
| $T$ | 3.04 | 2.04 | 2.05 | 2.73 |
| $T_b$ | | | 1.12 | 0.51 |

At inference time, ReBEAR employs the learned Gaussian policy, enabling efficient action sampling comparable to standard non-diffusion-based offline RL methods. To quantify this, we benchmarked inference time per trajectory on the AntMaze-umaze task using a Tesla V100 GPU. ReBEAR achieves an average inference time of 1.1 seconds per trajectory, whereas DQL—which directly relies on a diffusion model for action generation—takes 3.4 seconds. This confirms that ReBEAR maintains efficient inference.

## G  LIMITATIONS

ReBEAR offers efficient inference, as the learned policy is actually a Gaussian distribution. However, computing MMD requires sampling from the diffusion model, which incurs additional computational overhead. We leave the acceleration of denoising processes for training efficiency to future research.

## H  THE USE OF LARGE LANGUAGE MODELS

Large language models (LLMs) were used strictly for linguistic refinement of this manuscript, including correcting typos, checking grammar, and improving sentence fluency. No part of the technical content was generated by LLMs.

