# OpenReview forum: "Revisiting Maximum Mean Discrepancy via Diffusion Behavior Policy in Offline RL: A Mode-Seeking Perspective"
_ICLR.cc/2026/Conference — ICLR 2026 Conference Withdrawn Submission_

### Official Review · Reviewer_Fc2j · 2025-10-29

**Soundness:** 2
**Presentation:** 3
**Contribution:** 2
**Rating:** 4
**Confidence:** 4

**Summary:**

This paper revisits the use of Maximum Mean Discrepancy (MMD) as a policy constraint in offline reinforcement learning (RL) to address distributional shift. The authors' core argument is that prior $MMD$-based methods (like BEAR) failed not because $MMD$ is an unsuitable objective, but because they used low-fidelity behavior models (e.g., CVAE) that "blur" the underlying data distribution.
The paper presents two main contributions:
1. It demonstrates empirically (via 2D bandit tasks) and in D4RL benchmarks that $MMD$, when paired with a high-fidelity diffusion-based behavior policy ($Diff_{\omega}$), exhibits strong "mode-seeking" behavior. This allows the learned policy to focus on high-reward regions of the behavior policy.
2. It identifies that this mode-seeking capability is highly sensitive to Q-value overestimation. To mitigate this, the paper introduces a novel target Q-value penalty, $\mathcal{T}_{\textrm{MMD}}^{\pi} Q$, which penalizes the target value if the policy's next-state actions (under $\pi$) are far from the behavior policy (under $Diff_{\omega}$), as measured by $MMD$.

The resulting algorithm, "ReBEAR," is shown to significantly outperform the original BEAR algorithm and achieve competitive results against other state-of-the-art policy constraint methods on the D4RL benchmark.

**Strengths:**

1. **Excellent Diagnosis**: The primary strength is the clear and well-supported diagnosis of why prior $MMD$-based methods like BEAR failed. The insight that the CVAE's modeling error was the main issue, rather than the $MMD$ metric itself, is valuable.
2. **Strong Intuitive Visualizations**: Figure 1 is highly effective, providing a simple, clear visual proof-of-concept for the paper's central hypothesis.
3. **Significant Empirical Improvement (over BEAR)**: The paper shows a massive 103.2% performance gain over BEAR on MuJoCo, decisively proving its hypothesis.

**Weaknesses:**

1. **Flawed Core Premise**: The paper's entire motivation for using $MMD$ rests on a flawed premise. The abstract states: "Diffusion models... cannot be directly employed with KL divergence due to the absence of analytic probability formula." This assertion is incorrect. Concurrent work [Gao et al., 2025] provides a direct counterexample by deriving and optimizing an analytic, path-wise reverse $D_{KL}$ divergence. This fundamentally undermines the paper's argument for why $MMD$ must be used.
2. **Performance Claims are Not SOTA**: The paper claims "state-of-the-art performance." However, its total score on the 9 D4RL locomotion tasks (835.8) is outperformed by the BDPO algorithm (856.7), which itself is already surpassed by other recent work. Therefore, the performance is "competitive," but not SOTA.
3. **Weaker Divergence Metric**: Given that reverse $D_{KL}$ is known to be the correct "mode-seeking" divergence for this type of constrained optimization, and that it is tractable (per BDPO), the paper is left advocating for $MMD$, which is generally considered a weaker divergence metric. The paper does not provide a compelling reason (e.g., computational speed, stability) to prefer $MMD$ over the tractable $D_{KL}$ alternative.
4. **Theoretically Flawed Critic Penalty**: The $MMD$-based critic penalty (Eq. 7) is theoretically unsound as an OOD detector, as it fails in two ways: $MMD$ measures the global discrepancy between two distributions ($\pi(\cdot|s')$ and $Diff_{\omega}(\cdot|s')$), not the likelihood of specific actions being OOD.
    - False Positives (Penalizing $ID$ actions): If the behavior policy $Diff_{\omega}$ is multi-modal (e.g., modes A and B) and the learned policy $\pi$ (a single Gaussian) perfectly covers only Mode A, this policy is entirely in-distribution. However, $MMD(Diff_{\omega}, \pi)$ will be very large (due to the uncovered Mode B), causing the critic to incorrectly penalize this valid policy.
    - False Negatives (Failing to penalize $OOD$ actions): Conversely, and more critically, if the policy $\pi$ moves to a truly OOD region (e.g., the geometric center between modes A and B), $MMD(Diff_{\omega}, \pi)$ could be deceptively small as it "averages" the discrepancy to all modes. This means the penalty fails to trigger when it is needed most. The penalty seems to work heuristically by penalizing "shape mismatch," not by reliably identifying OOD actions.

**References**
Chen-Xiao Gao, Chenyang Wu, Mingjun Cao, Chenjun Xiao, Yang Yu, Zongzhang Zhang Proceedings of the 42nd International Conference on Machine Learning, PMLR 267:18630-18657, 2025.

**Questions:**

1. The paper's core premise rests on the intractability of $D_{KL}$ for diffusion models. How do the authors respond to the findings of Gao et al. (2025, "Behavior-Regularized Diffusion Policy Optimization"), which successfully derives and optimizes an analytic, path-wise reverse $D_{KL}$ regularizer for diffusion policies?
2. Given that this $D_{KL}$ approach is tractable and appears to achieve stronger empirical results (856.7 vs 835.8 on MuJoCo), what is the primary argument that remains for preferring the $MMD$-based approach of ReBEAR? Does $MMD$ offer significant advantages in computational cost, training stability, or ease of implementation that would make it a preferable choice over the $D_{KL}$ method?
3. The ablation for the critic penalty $\beta_c$ was very effective. However, its design based on $MMD$ seems theoretically flawed. Why did the authors choose this heuristic over a more standard, point-wise penalty, like a CQL-style term? Did the authors compare this $MMD$ penalty to a CQL-style penalty within their framework?

---

### Official Review · Reviewer_VFNn · 2025-10-30

**Soundness:** 2
**Presentation:** 1
**Contribution:** 2
**Rating:** 2
**Confidence:** 3

**Summary:**

The authors revisit an offline RL method (BEAR) that constraints the learned policy using a maximum mean discrepancy constraint to mitigate distribution shift.
They claim to achieve strong results, but I struggle to elicit precisely whereing their methodological contribution lies.

**Strengths:**

- Offline RL is an important topic with broad applications,
- The claimed results appear strong, though I suspect they may not be as strong as purported (see W1).

**Weaknesses:**

**W1. Restricted scope while making excessive claims**

The scope and positioning of the paper is primarily restricted to other policy constraint methods. It does not adequately compare and relate to other offline RL methods, such as ones based on Q-ensembles. Similarly, it mixes overstated claims of state-of-the-art results (e.g. "our method consistently achieves state-of-the-art performance across all tasks.", l.348;  "revolutionize the MMD-based policy
constraint method for offline RL.", l.27) with the caveat that "Our main comparisons focus on policy constraint methods" (l.308).

**W2. Introduction in need of revision**

The introduction is in need of a thorough revision. I would suggest the following:
- Introduce and explain policy constraint methods early, making it clear that is the main focus of the paper.
- Rewrite the second and third paragraphs (l.42-91) by synthesizing the related work into a coherent story. It currently reads too much like a shopping list (authors_1 did X, authors_2 did Y, etc.)
- Paragraph four (l.93-103) together with Figure 1 are key, as they're supposed to highlight the shortcoming of previous methods and explain the gist of the new idea. However, I think they are currently not clear at all and need to be simplified to become more broadly accessible. In particular, reduce the usage of jargon like "mode-seeking" and "behavior model" in favor of generally accepted terminology (defining things as necessary). In Figure 1, I think it's enough to have one of the two examples to get the point across, and reconsider whether you need all of the subfigures (many of them are not even defined here).
- You mention reverse KL constraints and their limitations in multiple places, but I struggle to see how that relates to the main discussion of the paper.
- I struggle to understand the last sentence: "ReBEAR can be viewed as distilling a Gaussian policy from the expressive diffusion-based behavior policy via MMD, which not only preserves the fidelity of the diffusion model but also enables efficient inference through the Gaussian model."

**W3. Grammatical errors and mathematical imprecision**

In multiple places, the grammar is incorrect or the text would benefit from language editing to improve clarity and flow. Some examples:
- l.53 "avoiding overly constraints on suboptimal regions",
- l. 135 "while restrict the learning policy",
- l.183 "CVAE tends to blur the support of the behavior policy". Technically, I think it's the feasible set and *not* the support you mean (this comment applies to many places where you're discussing the support).
- Figure 3 caption: "Therefore, mode-seeking property is demonstrated."
- Theorem 3.1, uses different notation from other parts of the text.

There are good tools available for this (including LLMs) and I encourage the authors to use them.

**Minor things:**
- l.187-201 The diffusion model seems pretty standard, hence I don't think there is a need to explain it in such detail unless you're somehow making an important modification.
- Figures 4 and 6: consider changing the limits on the y-axes to make the differences across methods easier to appreciate.

**Questions:**

1. Could you isolate the effects you highlight in the introduction (related to Figure 1) to show *quantitatively*, through well-designed experiments, that this effect also arises in more realistic problems?
2. Related to point 1, can you show that the CVAE in BEAR is properly trained? Based on Figure 1, it looks like it may not be (but I'm not 100% sure of what I'd expect to see, cf. W2).
3. How do you actually enforce the constraint in eq. (3)?
4. In the theoretical analysis in section 3.4, are your results restricted to Gaussian mixture distributions?

---

### Official Review · Reviewer_5kCQ · 2025-10-31

**Soundness:** 3
**Presentation:** 2
**Contribution:** 2
**Rating:** 4
**Confidence:** 3

**Summary:**

This paper proposes ReBEAR, a method that constraints the policy update using an MMD measure between the current policy and the behavior policy. Importantly, the behavior policy is modeled using a diffusion model as compared to the BEAR method. Additionally, the paper proposes using an MMD-based penalty for updating the Q-values that serves as a regularization to avoid valuing sub-optimal actions too high. The effectiveness of the method is demonstrated on several simulated tasks.

**Strengths:**

- The tackled problem in this paper is interesting and relevant for the RL community

**Weaknesses:**

- Several hyperparameters need to be tuned per task

- I find the presentation of the paper could be improved, and specific parts of the method can be motivated more clearly (also see questions).

**Questions:**

- Figure 1 shows that ReBEAR can focus on a single mode only (e.g. second and last row in Fig. 1). However, one can also see that this specific mode has the highest reward. What happens if two modes with the same reward, or very similar rewards, are present? Right now, there is a clear difference in the values of each mode.

- How is the actually optimized policy modeled? Is it a Gaussian? If it is a Gaussian, is it even possible that the optimized policy can represent multiple modes, which confuses me in the sense of why the target q-value penalty is needed (Section 3.2), as a Gaussian can only represent one mode?

- If it is not a Gaussian, is there a reason for not choosing a Gaussian policy representation?

- The paper states that evaluating the KL under diffusion models is not applicable. However, there exist variational approaches that have applied optimizing a diffusion policy using the reverse KL, notably in the online RL case [1], or distilling a Gaussian Mixture model from a diffusion behavior model [2] in the offline case. Especially the latter is a related approach, where the mixture model could easily be replaced by a Gaussian policy.

- The y-axis in Fig.2 is wrong (negative values)

- The idea of weighting the optimization with the Q-values has also been explored in advantage weighted regression (see e.g. [3,4,5]). It would be interesting to know whether this approach has also been considered for the mode-seeking behavior.

- Section 3.2 introduces an MMD-based operator to avoid sampling from a mode that has very similar rewards (Fig. 2). The MMD penalty on top of the Q-value is supposed to avoid sampling from this other mode that can occasionally lead to higher rewards due to noisy rewards, although the mean is smaller than the targeted mode. However, the penalty term is based on the diffusion behavior policy, which actually could also generate samples from this disliked mode. Therefore, I don't really see why this penalty term avoids sampling points from the noisy mode. If the behavior policy has support for the other mode, then the MMD should also be small, which also leads to no punishment. How is the opposite enforced?

- I appreciate the analysis on the sensitivity of beta_c. However, I find Fig. 5 rather confusing as the same colors represent different values for beta_c in the curves. I would recommend choosing a set of beta_c values with the same colors for all environments to have a consistent comparison.

- How many samples are required for a reliable MMD estimate? From my understanding, the MMD is based on samples that use kernels to evaluate the "distance". I understand that the more samples the better the MMD, but in this case, it is also required to evaluate the kernels for each of the samples, which is, I assume, a computational bottleneck?

[1] O. Celik, et al. DIME: Diffusion-Based Maximum Entropy Reinforcement Learning. ICML 2025.

[2] H. Zhou, et al. Variational Distillation of Diffusion Policies into Mixture of Experts. NeurIPS 2024.

[3] J. Peters, et al. Reinforcement Learning by reward-weighted regression for operational space control. ICML 2007.

[4] X.B. Peng, et al. Advantage-Weighted Regression: Simple and scalable off-policy Reinforcement learning. arxiv 2019

[5] A. Nair, et al. AWAC: Accelerating online reinforcement learning with offline datasets. arxiv 2021.

---

### Official Review · Reviewer_YF7U · 2025-11-01

**Soundness:** 3
**Presentation:** 2
**Contribution:** 1
**Rating:** 2
**Confidence:** 4

**Summary:**

This paper revisits the role of Maximum Mean Discrepancy (MMD) as a policy constraint in offline reinforcement learning (RL). The authors argue that prior MMD-based methods (e.g., BEAR) failed mainly due to distorted behavior modeling rather than MMD itself. They combine diffusion-based behavior policies with MMD to exploit its mode-seeking property and further introduce a target Q-value penalty to counteract value overestimation. The proposed method, ReBEAR, significantly outperforms BEAR and other policy constraint baselines on the D4RL benchmark, achieving strong state-of-the-art results.

**Strengths:**

1. The paper is overall well-written.
2. It is meaningful to explore the performance of the diffusion model with the previous offline policy constraint.

**Weaknesses:**

1. The novelty and contribution of this paper seem limited, which seems to just replace the Gaussian policy with a diffusion model in BEAR.
2. The author does not illustrate the difficulty of incorporating the MMD constraint into the diffusion policy.
3. The reviewer suggests adding an algorithm box to depict the pipeline of the algorithm straightforwardly.
4. In Figure 1, it is a little strange that ReBEAR does not show the multimodality of the diffusion model.

**Questions:**

1. For the diffusion model, it seems time-consuming to compute the sample-based MMD constraint, especially to do the subsequent backpropagation. How can we avoid that?
2. Does Figure 1 indicate that ReBEAR can suppress the multimodality of the diffusion model? In that case, what is the advantage of using a diffusion model to replace a Gaussian policy in BEAR?

---

### Note · Authors · 2025-11-12

I have read and agree with the venue's withdrawal policy on behalf of myself and my co-authors.